# TLSCA-SVM Fault Diagnosis Optimization Method Based on Transfer Learning

**Aihua Zhang** [1,*] , **Danlu Yu** [2] **and Zhiqiang Zhang** [1]

1 College of Physical Science and Technology, Bohai University, Jinzhou 121013, China; zhangzhiqiang@qymail.bhu.edu.cn
2 College of Control Science and Engineering, Bohai University, Jinzhou 121013, China; yudanlu15178672625@163.com
* Correspondence: jsxinxi_zah@163.com; Tel.: +86-138-4063-8593

**Abstract:** In fault-diagnosis classification, a pressing issue is the lack of target-fault samples. Obtaining fault data requires a great amount of time, energy and financial resources. These factors affect the accuracy of diagnosis. To address this problem, a novel fault-diagnosis-classification optimization method, namely TLSCA-SVM, which combines the sine cosine algorithm and support vector machine (SCA-SVM) with transfer learning, is proposed here. Considering the availability of fault data, this thesis uses the data generated by analog circuits from different faults for analysis. Firstly, the data signal is collected from different faults of the analog circuit, and then the characteristic data are extracted from the data signals by the wavelet packets. Secondly, to employ the principal component analysis (PCA) reduces the feature-value dimension. Lastly, as an auxiliary condition, the error-penalty item is added to the objective function of the SCA-SVM classifier to construct an innovative fault-diagnosis model namely TLSCA-SVM. Among them, the Sallen–Key bandpass filter circuit and the CSTV filter circuit are used to provide the data for horizontal- and vertical-contrast classification results. Comparing the SCA with the five optimization algorithms, it is concluded that the performance of SCA optimization parameters has certain advantages in the classification accuracy and speed. Additionally, to prove the superiority of the SCA-SVM classification algorithm, the five classification algorithms are compared with the SCA-SVM algorithm. Simulation results showed that the SCA-SVM classification has higher precision and a faster response time compared to the others. After adding the error penalty term to SCA-SVM, TLSCA-SVM requires fewer fault samples to process fault diagnosis. Ultimately, the method which is proposed could not only perform fault diagnosis effectively and quickly, but also could run effectively to achieve the effect of transfer learning in the case of less failure data.

**Keywords:** TLSCA-SVM; optimization method; fault diagnosis; transfer learning

## 1. Introduction

In terms of practical application, fault diagnosis is primarily used in industrial failure. Among such incidents, 80% of industrial failures come from analog circuits. Therefore, analog circuit fault diagnosis is the research focus of industrial fault diagnosis. Compared with digital circuits, analog circuits are more complicated and have more stringent requirements for fault diagnosis. Training using analog circuits for diagnostic models in fault diagnosis is more valuable. With the continuous refinement of various data-analysis methods, the troubleshooting technology for analog circuits is also improving; common troubleshooting techniques, including PCA [1], Search Grid [2], particle swarm optimization (PSO) [3], ant colony algorithm (ACA) [4], simulated annealing (SA) [5], genetic algorithm (GA) [6], Back Propagation Neural Network (BP) [6], Self-organizing Maps (SOM) [7], Extreme Learning Machine (ELM) [8], decision tree [9], random forest [10] and SVM [11] all have good classification results to some extent [12].

For the fault diagnosis of analog circuits [13], the following operations are required. Firstly, an analog circuit model is built, the fault is set in the analog circuit and the failure data is collected. Secondly, the fault data is processed, using methods such as feature-value extraction, dimensionality reduction, linearization, etc. Finally, the processed data is used in the modelling of fault diagnosis to judge the performance of the model [14]. In this article, the data of Sallen–Key circuit is used for model comparison and the more complex CSTV circuit is used for validation to determine the versatility of the model. Here, the data of both analog circuits are used for horizontal and vertical comparison experiments of the algorithm in order to obtain more practical conclusions. When there are fewer fault samples, the auxiliary condition, namely the error penalty item, is added to the objective function of the SCA-SVM [15] classifier to construct a new fault diagnosis model, which is TLSCA-SVM. The model not only has the advantages of the SCA-SVM fault diagnosis model, i.e., the fault classification speed is fast, the accuracy is higher, but also the transfer-learning [16] ability is better.

As a new concept of machine learning [17], transfer learning brings new research directions to machine learning. This paper uses the data from different faults in the analog circuit for fault diagnosis [18], performs a series of data processing, combines the SCA optimization method with SVM to obtain the classifier SCA-SVM and adds an error penalty item to build a new fault diagnosis model, namely the TLSCA- SVM model. This method imports the data processed by the wavelet packet and PCA into the improved SCA-SVM classifier for training and prediction, thereby improving the speed and accuracy of diagnosis [19].

The controversy about the SCA algorithm [20] concerns the local search of the optimization algorithm. It is not difficult to understand that the optimal solution can be guaranteed using global search [21]. However, it is inevitable that the local optimal solution may appear in the local search, performance at a certain point is best, and the farther away from the point, the worse the performance. In response to this debate, this paper effectively avoids the occurrence of the problem. In this paper, the parameters of SVM are optimized by using the same characteristics of local search and global search probability of the SCA algorithm in optimizing parameters. The SCA-SVM classifier and a good classification effect are obtained.

As machine learning continues to innovate, the concept of transfer learning has been introduced through research. Traditional machine learning uses the continuous autonomous acquisition of knowledge from the data. This mechanism requires a lot of data and iterations to achieve data-driven effects. In machine learning, when a trained model is applied to a new field, the effect is often poor. Transfer learning, as an important branch of machine learning, focuses on applying knowledge that has already been learned to new problems, so that the knowledge transfer can allow trained models to be applied to new areas. Transfer learning includes zero-shot learning [22], one-shot learning [23] and few-shot learning [24]. Since the zero-shot learning is too ideal to exist in actual data processing, the limit of few-shot learning is zero-shot learning. Therefore, in this paper about analog circuits fault diagnosis, few-shot learning is used for fault diagnosis [25], as well as transfer learning, in order to achieve the effect of fault prediction [26].

Transfer learning refers to the transfer of knowledge from one field to another, and it is common to apply the experience gained from learning to new learning, which is the influence of one learning state on another. The earliest application of transfer learning was in educational psychology, which put forward that the implementation of transfer learning must combine old with new knowledge. In terms of machine learning, transfer-learning development has a better development prospect. Since 1995, NIPS launched a "Learning to Learn" professional seminar in the field of machine learning [27], and since then transfer-learning research has entered a period of rapid development. Transfer learning is to apply the knowledge extracted from the source task [28] to the target task. Compared with multitasking, the most important thing in transfer learning is the target task [29], rather than extracting knowledge of all source tasks at the same time. Of course, in educational psychology, the implementation object of transfer learning is human, while

the implementation object in machine learning is the corresponding algorithm model. In this regard, humans have excellent transfer-learning capabilities. For example, in human childhood, when young children are provided with oral descriptions by their parents, the children can find the corresponding objects. Even in the learning process, as long as the solution is mastered, similar problems can be solved. However, the application of transfer learning in machine learning is not particularly easy. This is because, unlike humans, machines cannot achieve the so-called independent thinking and can only achieve the desired results through continuous data optimization. It is worth mentioning that in the face of massive data analysis, the introduction of migration learning can reduce a lot of the unnecessary workload, which has certain research significance.

At present, there are many scholars focused on transfer learning research and most people research the algorithm related to transfer learning. They use different technologies to achieve the requirement of transfer learning. Many transfer-learning ideas are applied to image recognition [30], such as image features being extracted to a training set, the knowledge of the source task is transferred to the target task, and finally the image recognition in the test set can be realized.

This article uses transfer learning for the purpose of fault diagnosis. Obtaining fault data in industrial fault diagnosis requires a lot of manpower and material resources. The fault of actual industrial equipment is a kind of damage. Therefore, the application of fault-diagnosis methods in transfer learning is typical. The use of normal data and a some fault data can effectively diagnose faults and achieve predictive effects. This method can maintain equipment safety and ensure the normal operation of actual industrial equipment. Considering the availability of fault data, this paper uses the data generated by analog circuits from different faults for analysis. An error penalty is added to the SCA-SVM classifier to build an innovative fault-diagnosis model, which is TLSCA-SVM. By comparing the experiments, it is concluded that the model can diagnose the fault with less fault data, achieve the effect of transfer learning, and finally realize the prediction of failure.

## 2. SCA-SVM Algorithm

The SCA optimization parameter algorithm is a novel random optimization method based on population. The SCA was proposed by Australian scholar Mirjalili in 2016 [31], the essence of which is to optimize the parameter by using random probability. SCA creates randomly generating multiple initial candidate solutions, and uses sine and cosine functions to make these initial candidate solutions have the same probability of moving either in the direction of the optimal solution or reverse. This method not only guarantees the accuracy of global search optimization parameters, but also ensures the speed of local search optimization parameters, so that the optimal parameters can be found quickly and accurately in the model. Because the essence of the SCA optimization algorithm is the population optimization algorithm, it also meets the general law of the two stages of the population optimization algorithm, which is exploration and utilization. When the algorithm is in the exploration stage, the algorithm randomly searches with a large probability gradient to ensure sufficient search space. When the algorithm is in the utilization stage, the random probability gradient decreases gradually, ensuring that the optimal solution can be found accurately. In SCA [32], the optimization process is divided into global search and local search. These two parts promote and restrain each other, as global search is used to quickly locate the optimal solution of the range, and local search is used to find the optimal solution. These two parts reach a dynamic balance can find the global optimal solution. If there is only a global search, the optimization speed is slow and the equipment configuration is high. If there is only a local search, it is easy to obtain a local optimal solution. The SCA optimization parameter algorithm is determined by the different value situation of the sine function global search and local search.

The two important parameters of SVM are kernel function [33] and the penalty factor [34]. These two parameters directly affect the performance of SVM [35], so finding the optimal parameters of SVM becomes the core problem of constructing a classification

model. How to find the optimal parameters quickly and effectively becomes the key to the optimization algorithm. This article introduces SCA to optimize the SVM parameters. SCA randomly creates multiple initial candidate solutions and uses sine and cosine functions [36] to determine the search method. This method ensures that the initial candidate solution moves toward or away from the optimal solution with the same probability. This way of finding the optimal solution not only ensures the accuracy of the global search optimization parameters, but also ensures the speed of the local search optimization parameters, so that the optimal parameters can be found quickly and accurately in the model [37].

### 2.1. Principles of SCA Optimization Parameters

In this essay, the SCA optimization parameter algorithm can be used to optimize two important parameters in SVM. The essence of the SCA algorithm is to use the unique characteristics of the trigonometric function to make the probability of each optimization the same, so that the optimization effect meets the accuracy of the global search and the speed of the local search. The specific operation is that the value of the sine and cosine function is used to determine whether to perform a global search. For example, when the distance between the value of the sine and cosine function and the abscissa exceeds 0.5, that is the function value is distributed between 0.5 and 1 or $-0.5$ and $-1$, the optimization of the global search method is achieved. When the distance from the sine and cosine function value to the horizontal coordinates is less than 0.5, which is the function value is distributed between $-0.5$ and 0.5, the SCA algorithm conducts a local search. SCA randomly creates multiple initial-candidate solutions and uses sine and cosine functions to move these initial candidates in the same probability of moving or reversing in the direction of the best solution, which not only ensures the accuracy of the global-search optimization parameters, but also ensures the speed of the local search [11].

In the SCA optimization algorithm, the trigonometric function determines the position of the next iteration point and is iterate according to the following formula:

$$X_i^{t+1} = X_i^t + \theta_1 + \sin\theta_2 + \left|\theta_3 P_i^t - X_i^t\right| \tag{1}$$

$$X_i^{t+1} = X_i^t + \theta_1 + \cos\theta_2 + \left|\theta_3 P_i^t - X_i^t\right| \tag{2}$$

In the equation, $X_i^t$ is the position of the current iteration, $\theta_1\theta_2$ and $\theta_3$ are random numbers of random values in iteration that decide the trigonometric function, $P_i^t$ is the best selected scenario. In the actual optimization process, the above positions are selected appropriately and the equation can be updated with the sine cosine function:

$$\begin{cases} X_i^{t+1}=X_i^t+\theta_1+\sin\theta_2+\left|\theta_3 P_i^t - X_i^t\right|, & \theta_4<0.5 \\ X_i^{t+1}=X_i^t+\theta_1+\cos\theta_2+\left|\theta_3 P_i^t - X_i^t\right|, & \theta_4\geq0.5 \end{cases} \tag{3}$$

In the equation, $\theta_4$ is a number randomly selected each time in the range of 0 and 1, ensuring that the probability is the same when optimizing.

In the above formula, according to parameter $\theta_1$, the algorithm can determine the direction of the next iteration. The operation occurs when the distance from the parameter value to the abscissa exceeds 0.5, that is, the value is distributed between 0.5 and 1 or between $-0.5$ and $-1$, in the outer ring area shown in Figure 1. When the distance between the parameter value and the abscissa is less than 0.5, that is, the function value is distributed between $-0.5$ and 0.5, the search is performed in the circle area shown in Figure 1. The parameter is a number randomly selected between 0 and $\pi$ each time. Parameter $\theta_3$ is used to determine the random allocation of the enhanced or weakened relationship, and a size comparison with 1 has been used to determine whether the relationship is enhanced or weakened. As can be seen from the Formula (3), the parameter value of $\theta_4$ can select the type of the sine and cosine function, when $\theta_4 < 0.5$, the equation containing only the sine function is selected for the next iteration. When $\theta_4 \geq 0.5$, the equation containing only the cosine function is selected for the next iteration. Through the above iteration, it can be guaranteed that the algorithm has the same probability in global search and local search,

so that the algorithm itself can combine the benefits of both search modes itself to achieve better results.

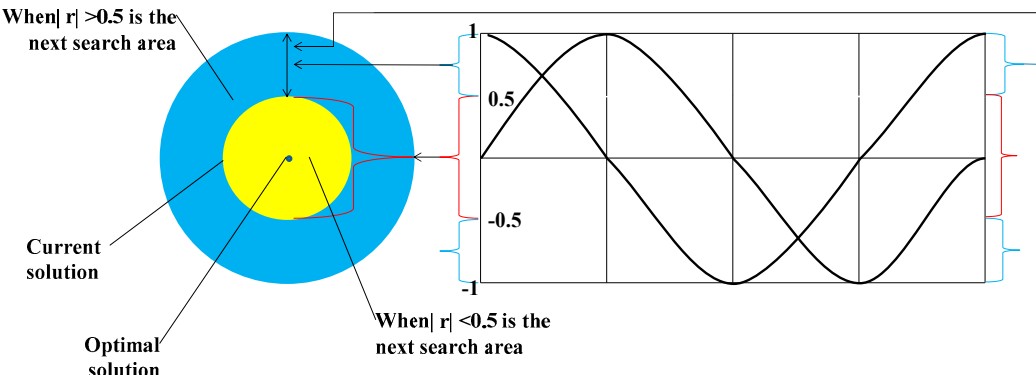

**Figure 1.** SCA optimization algorithm iterative schematic.

In order to satisfy the global search and the local search with the same probability conditions, and make the optimization solution converge, the SCA optimization algorithm self-adjusts using Formula (4):

$$\theta_1 = a - t\frac{a}{T} \tag{4}$$

### 2.2. The Classification Principle of SVM

SVM was originally used to deal with dichotomous problems. In order to meet more classification problems, slack variables and maximum separation hyperplane theory were continuously introduced and applied to classify nonlinear problems. Due to the special theoretical concept, the SVM classifier effect can be applied to transfer learning, so this paper used the SVM for fault diagnosis based on transfer learning.

The most important problem of SVM is the mapping of kernel functions. It is a nonlinear indistinguishable problem. It is classified by hyperplane and mapped to a simple linear problem, where the largest hyperplane satisfies the following conditions:

$$\begin{cases} \min\phi(\omega)=\frac{1}{2}\|\omega\|^2 \\ \text{s.t.}\, y_i(\omega^T x+b)\geq 1, i=1,2,3,\ldots,n \end{cases} \tag{5}$$

$\omega$ is the general vector of the maximum hyperplane, and *b* is the offset of the maximum hyperplane, parameter $\xi$ ($\xi > 0$) determines the boundaries.

$$y_i[\omega \cdot \phi(x_i) + b] + \xi_i \geq 1, i = 1, 2, \ldots, n \tag{6}$$

The penalty factor C is applied to alter the proportion of relaxation variable $\xi_i$, and Formula (5) is altered to:

$$\begin{cases} \min(\omega)=\frac{1}{2}\|\omega\|^2+C\sum_{i=1}^{n}\xi_i \\ y_i[\omega\cdot\phi(x_i)+b]+\xi_i\geq 1, i=1,2,\ldots,n \end{cases} \tag{7}$$

By introducing Lagrange multiplier $\alpha$, the maximum hyperplane problem as a pair problem is reflected as follows:

$$\begin{cases} \max L(\alpha)=\sum_{i=1}^{n}\alpha_i+\frac{1}{2}\sum_{i=1}^{n}\sum_{j=1}^{n}\alpha_i\alpha_j y_i y_j K(x_i\cdot x_j) \\ \text{s.t.}\sum_{i=1}^{n}\alpha_i y_i=0, 0\leq\alpha_i\leq C, i=1,2,\ldots,n \end{cases} \tag{8}$$

Solving the upper class, the decision function of the SVM is shown below:

$$f(\mathbf{x}) = \mathrm{sgn}\left[\sum_{i=1}^{n} \alpha_i \mathbf{y}_i K(\mathbf{x}_i \cdot \mathbf{x}_j) + b\right] \tag{9}$$

In this paper, the important parameters penalty factor C and kernel parameter in SVM were perfected by SCA optimization parameters, and a fault classifier with excellent classification performance is obtained.

## 3. Fault Diagnosis Model of TLSCA-SVM Algorithm

### 3.1. Principles of Transfer Learning

Transfer learning is the novel branch of machine learning, which has a critical influence on the data processing of artificial intelligence. In 1995, the NIPS professional seminar "Learning to Learn" was launched in the field of machine learning 26. Since then, transfer-learning research has entered a period of rapid development. The main principle of transfer learning is to apply the knowledge of the source task to the target task. Compared with multi-task learning, the most important thing in transfer learning is the target task, rather than learning the source task that applies to the target task.

Transfer learning mainly includes the following two concepts: domain $D$ and task $H$. The two parts of $D$ are composed of two parts: the feature space $E$ and the edge probability distribution $P(e)$, where $\{e_1, \ldots, e_n\} \in E$, for a given domain, $H$ also consists of two parts.

$D_s$ is given a labeled source domain data, $H_s$ is the corresponding source task. $D_t$ is a target domain with very few tags, $H_t$ is the corresponding learning task. The purpose is to transfer the knowledge of $D_s$ and $H_s$ to $D_t$, so as to perfect the performance of the target function ($D_s \neq D_t, H_s \neq H_t$).

### 3.2. TLSCA-SVM Algorithm

The SCA optimization algorithm uses the equality constraints to optimize the parameters of the SVM classifier. This method transforms the solution of the optimization problem into the solution of a set of linear equations. For a set of input samples $\{(d_i, g_i)\}, i = 1, \ldots, n, d_i \in R^n$. SVM maps the nonlinear inseparable problem to a high-dimensional space into a linearly separable problem through kernel function mapping, where the function discriminant is:

$$f(d) = u^T \varphi(d) + q \tag{10}$$

To propose the migration algorithm of the auxiliary data set, the auxiliary data sets are similar to the target set to enhance the accuracy of the classification model. Given that $\{(d_i, g_i)\}_{i=1}^{n}$ is the target sample data, $\{(d_i, g_i)\}_{i=n+1}^{n}$ is similar sample data. On the basis of the original SCA-SVM optimization problem, the TLSCA-SVM method adds auxiliary data that is an error penalty term, to the target formula to realize the transfer of knowledge. The improved objective function can be expressed as:

$$\begin{cases} \min J(u,\lambda) = \frac{1}{2} u^T u + \frac{\beta_t}{2} \sum_{i=1}^{n} \lambda^2 + \frac{\beta_s}{2} \sum_{i=n+1}^{n+m} \lambda_i^2 \\ s.t. g_i(u^T \varphi(d_i) + q) + 1 - \lambda_i, i=1,2,\ldots,n,\ldots,n+m \end{cases} \tag{11}$$

In the formula, $\beta_t$ and $\beta_s$ are the penalty parameters of the target domain and source domain data sets, respectively, $\lambda_i$ is the prediction error.

When the Lagrange multiplier is introduced, the dual equation of Equation (10) is derived, multiply each equation constraint with the Lagrange multiplier $\alpha_i$, Then, the Lagrange function is established by adding the objective function:

$$L(u, q, \lambda_i, \alpha_i) = J(u, \lambda) - \sum_{i=1}^{n+1} \alpha_i \left[g_i(u^T \varphi(d_i) + q) - 1 + \lambda_i\right] \tag{12}$$

The necessary condition for taking the extreme value is to find the partial derivative of each variable and set it to zero to obtain the Formula (13).

$$
\begin{cases}
\frac{\partial L}{\partial u} = 0 \Rightarrow u = \sum\limits_{i=1}^{n+m} \alpha_i g_i \varphi(d_i) \\
\frac{\partial L}{\partial \lambda_i} = 0 \Rightarrow \alpha_i = \begin{cases} \beta_t \lambda_i, i = 1, 2, \ldots, n \\ \beta_s \lambda_i, i = n+1, \ldots, n+m \end{cases} \\
\frac{\partial L}{\partial q} = 0 \Rightarrow \sum\limits_{i=1}^{n} \alpha_i g_i = 0 \\
\frac{\partial L}{\partial \alpha_i} = 0 \Rightarrow g_i(u^T \varphi(d_i) + q) - 1 + \lambda_i = 0
\end{cases}
\tag{13}
$$

Eliminate variables $u$ and $\lambda_i$ to get the matrix equation:

$$
\begin{bmatrix} \Omega + \frac{1}{\beta} & G \\ G^T & 0 \end{bmatrix} \begin{bmatrix} \alpha \\ q \end{bmatrix} = \begin{bmatrix} e \\ 0 \end{bmatrix}
\tag{14}
$$

In the formula, $\Omega = g_i g_j k(d_i, d_j)$, $k(d_i, d_j) = \varphi(d_i)^T \varphi(d_j)$ is the kernel function. $G = [g_1, \ldots, g_{n+m}]^T$ and $e = [1, \ldots, 1]^T$ are $n+m$ dimensional column vectors, $\beta = diag[\beta_t, \ldots, \beta_t, \beta_s, \ldots, \beta_s]$.

The expression of TLSCA-SVM decision function is obtained by Formula (14).

$$
G(d) = \text{sgn}\left[ \sum_{i=1}^{l} \alpha_i g_i \varphi(d_i)^T \varphi(d_i) + q \right]
\tag{15}
$$

The Gaussian kernel function has an excellent anti-interference ability to the noise of the data. In this paper, the Gaussian kernel is used as the kernel function of TLSCA-SVM. The expression is:

$$
k(d, d_i) = \exp\left( \frac{-\|d - d_i\|^2}{2\delta^2} \right)
\tag{16}
$$

where $\delta$ is the parameter of the Gaussian kernel width.

Finally, combined with the previous SCA-SVM algorithm, the decision function of the TLSCA-SVM classifier is obtained by the improved algorithm.

### 3.3. Fault Diagnosis Process of TLSCA-SVM Algorithm

In this paper, the TLSCA-SVM algorithm is applied to an analog circuit fault analysis, which is accomplished via the following main four steps (shown from step 1–step 4): building analog circuit model, setting and collecting fault data, using wavelets for data feature extraction, using PCA dimension reduction, using TLSCA-SVM classifier for troubleshooting, and comparing the actual fault data with the fault diagnosis results to arrive at the fault diagnosis rate.

The programming in this article is based on the mutual control of multiple modules. The data processing uses a five-layer wavelet packet and PCA technology. The classifier part uses SCA to perfect the parameters in the SVM to achieve fast and accurate classification results. When the classification algorithm adds auxiliary conditions, the algorithm has the ability of transfer learning. The specific fault handling process is shown in Figure 2.

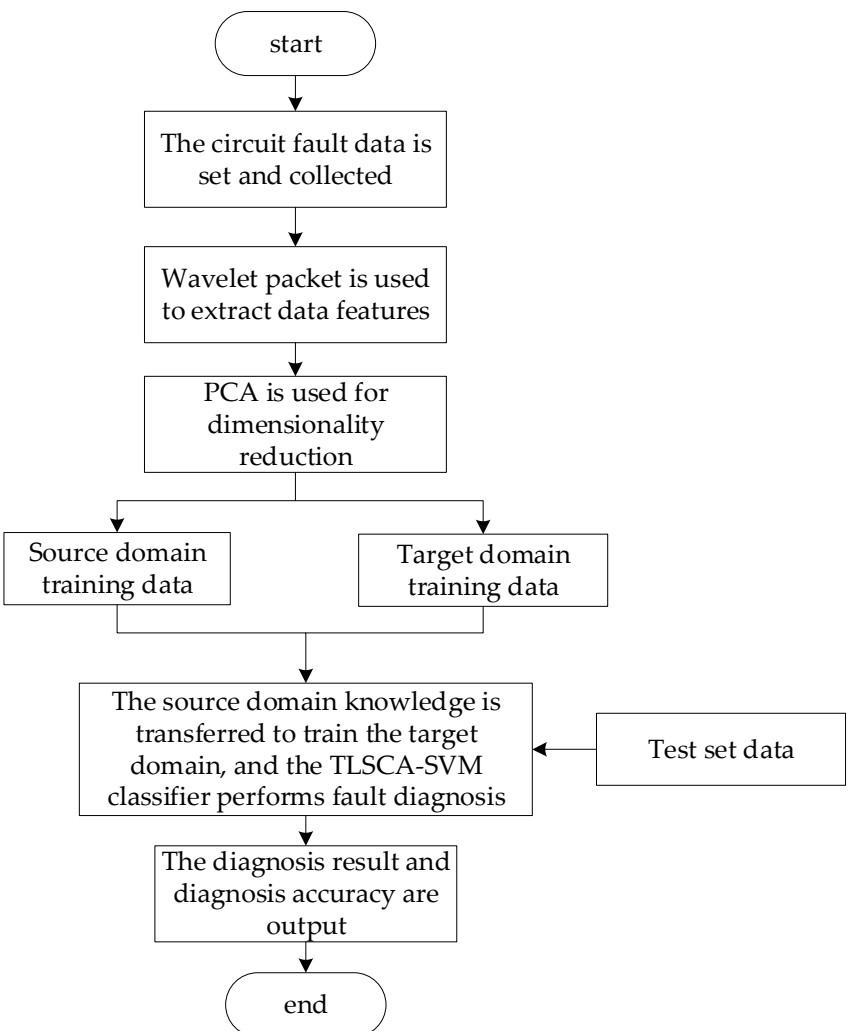

**Figure 2.** TLSCA-SVM fault diagnosis specific operation flowchart.

In this paper, the steps of fault diagnosis are shown below. The entire diagnostic process is not changed when comparing algorithms, but the comparison is made with the classifier and the optimized parameters of the classifier. This ensures that when comparing algorithms, the external conditions are consistent, and only the diagnostic effect of the classifier itself is considered. In order to ensure the consistency of the external conditions of the comparison algorithm, the same failure training set data is the same when using different classifiers for troubleshooting.

Step 1. The circuit model is established to collect and process data from different faults in the analog circuit. Input variables are obtained and the input data undergoes a series of data processing steps such as wavelet packet extraction, feature-value processing and PCA dimensionality reduction.

Step 2. After the above data processing is completed, the data is put into the pre-built classification model to achieve the effect of classifying the fault data. In this paper, the function of SVM is realized by LIBSVM. The penalty factor and kernel parameter in SVM are perfected by the SCA optimization method. In the SCA optimization method, multiple initial candidate solutions are randomly created. These initial candidates are used with the same probability of moving or reversing in the direction of the best solution. When looking for a relatively good solution, this method chooses to keep and proceed to the next iteration.

Step 3. An auxiliary condition is added to the objective function of the already debugged SCA-SVM classifier that is an error penalty term. An innovative fault diagnosis model, namely TLSCA-SVM, is constructed. The knowledge of the source domain is trans-

ferred to train the target domain, and the TLSCA-SVM classifier performs fault diagnosis. Fewer fault data is used as a training set and put into the classifier for fault diagnosis

Step 4. The trained TLSCA-SVM classifier model is used to classify faults on the test set, output the diagnosis results and judge the accuracy of the fault diagnosis results.

In this article, it is worth mentioning that the problem of a single-fault diagnosis in analog circuits is considered. Single fault refers to changing the parameters of only one component in the circuit while the parameters of other components in the circuit remain unchanged. The situation in which multiple component parameters are changed at the same time is referred to as a multiple-fault diagnosis problem. The data processing method of multi-fault diagnosis is similar to that of single-fault diagnosis. Multiple faults are changing the parameters of only two or more components while the parameters of other circuit components remain unchanged. Single fault is the basis of fault diagnosis in analog circuit fault diagnoses. In the actual analog circuit fault, the occurrence probability of a single fault is more than 80%, and the occurrence probability of multiple faults is relatively low. Additionally, a multi-fault diagnosis can be regarded as multiple single-fault problems occurring at the same time. That is, a multi-fault problem can be decomposed into multiple single-fault problems. Single-fault diagnosis and multi-fault diagnosis have many similarities in analog circuit fault diagnosis and multi-fault problems with less probability can be decomposed into single-fault for processing. Considering all of these aspects, this paper does not analyze the multi-fault diagnosis.

## 4. Acquisition and Process Fault Samples of Analog Circuits

This article takes the Sallen–Key circuit [38] and CSTV circuit [39] as diagnostic examples. Sallen–Key circuits and CSTV circuits are typical circuits that are often used to analog-circuit fault diagnosis. In the simulation experiment, considering the selectivity of the simulated circuit data, for more rigorous consideration, the faults of the Sallen–Key circuit and the CSTV circuit were set according to the literature [40–42].

At the same time, the Sallen–Key circuit is used because it is relatively simple as a second-order circuit and can use a public dataset. The Sallen–Key circuit is widely used in the fault diagnosis of analog circuits, so the Sallen–Key circuit was used to verify the validity of the algorithm. The CSTV circuit is a fourth-order circuit, which is more complex than the Sallen–Key circuit. The application of this circuit can show that the algorithm itself has universal applicability

### 4.1. Data Processing of Sallen-Key Band-Pass Filter Circuit with Injected Fault

In this essay, the data were studied by extracting the data in different modes of the Sallen–Key circuit. Circuit structure, individual component types and values are shown in Figure 3. Here, the parameter value is set to deviate from the original value by 50%, the capacity is set to deviate from the original value by 10%, and the resistance is set to deviate from the original value by 5%.

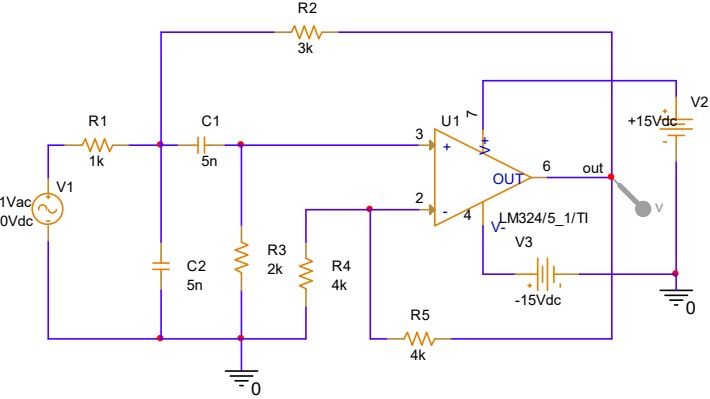

**Figure 3.** Sallen-Key band-pass filter circuit.

The excitation-signal parameters of the analog circuit are shown in Figure 3. After sensitivity analysis, R2, R3, C1, and C2 are selected as faulty components. The fault type and parameter list are shown in Table 1. The data of Sallen−Key circuit contains eight failure modes, which includes R2+, R2−, R3+, R3−, C1+, C1−, C2+ and C2−. The symbols − and + denote additional or little data and normal state (no failure, NF).

**Table 1.** Fault type and encoding of Sallen-Key band-pass filter circuit.

| Malfunction Coding | Failure Mode | Nominal Value | Fault Value |
|---|---|---|---|
| F0 | NF | - | - |
| F1 | R2+ | 3 KΩ | 4.5 KΩ |
| F2 | R2− | 3 KΩ | 1.5 KΩ |
| F3 | R3+ | 2 KΩ | 3 KΩ |
| F4 | R3− | 2 KΩ | 1 KΩ |
| F5 | C1+ | 5 nF | 7.5 nF |
| F6 | C1− | 5 nF | 2.5 nF |
| F7 | C2+ | 5 nF | 7.5 nF |
| F8 | C2− | 5 nF | 2.5 nF |

In this paper, the data of the analog circuit output signal waveform is collected, and the nine modes are set, which are the signal data of the normal state and the signal data of the eight fault states. Each mode collects 100 sample points, so there are a total of 900 state sample points, which are randomly selected. A total of 600 sample points were used as the training set, and the remaining 300 sample points were used as the test set.

### 4.2. Data Processing of the CSTV Filter Circuit Injected into the Fault

To reflect the generality of the SCA-SVM classifier to fault diagnosis, a relatively complex circuit, CSTV filter, was selected as the verification analog circuit. The circuit element name and value are shown in Figure 4, and the status type is shown in Table 2. A total of 1800 samples, 900 of which were collected as the training set, and the other 900 samples were used as the test set.

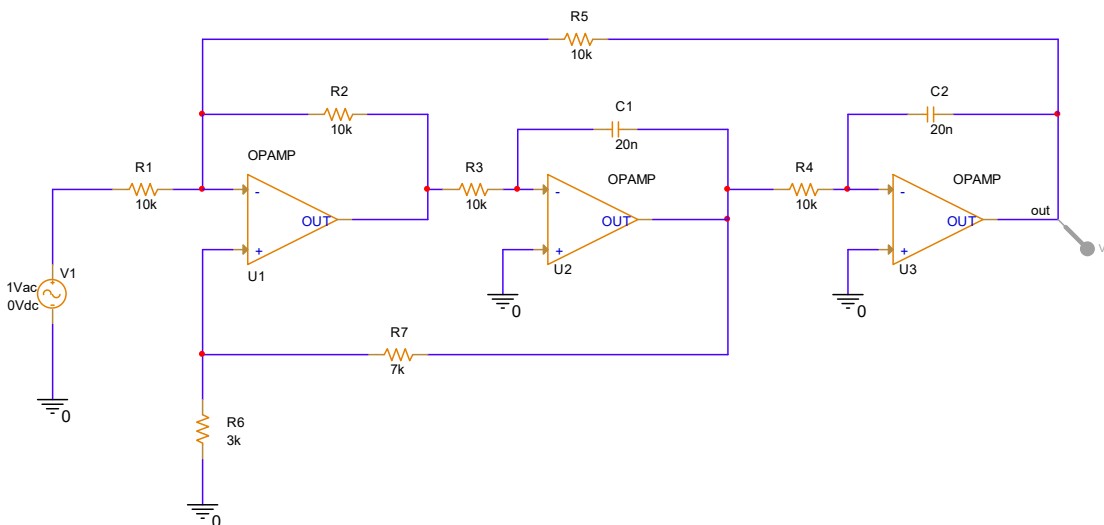

**Figure 4.** CSTV filter circuit.

The specific operation is to compare the classification algorithm and optimize the parameter method. The common second-order filter Sallen-Key circuit is applied for comparison experiments. To verify the versatility of the comparison algorithm conclusions, the multi-stage filter, which is a CSTV filter, is used. As verification, a more general result can be obtained.

Because the experimental simulation environment and experimental data are different, the experimental results will also be different. Therefore, when simulating the soft fault of

the circuit, a more general experimental environment and data set were selected. According to the above parameter settings, each of the nine modes are analyzed using 100 MC, the start frequency is set to 100 Hz, and the cutoff frequency is set to 10k Hz. A voltage probe is set at the output node of the circuit to collect the output voltage. This ensures that the experimental results studied in this paper are more extensive and reliable.

**Table 2.** Fault type and encoding of CSTV filter circuit.

| Malfunction Coding | Failure Mode | Nominal Value | Fault Value |
|---|---|---|---|
| F0 | NF | - | - |
| F1 | R1+ | 10 KΩ | 15 KΩ |
| F2 | R1− | 10 KΩ | 5 KΩ |
| F3 | R2+ | 10 KΩ | 15 KΩ |
| F4 | R2− | 10 KΩ | 5 KΩ |
| F5 | C1+ | 20 nF | 30 nF |
| F6 | C1− | 20 nF | 10 nF |
| F7 | C2+ | 20 nF | 30 nF |
| F8 | C2− | 20 nF | 10 nF |

### 4.3. Feature Processing of Analog Circuit Fault Data

In this paper, the processing of analog circuit fault data is divided into the three steps, which includes the extraction of fault signals, wavelet packet extraction of data features and PCA dimensionality reduction.

#### 4.3.1. Extract Fault Signal

Multiple fault control groups were established, the original data set was simulated, a voltage probe was set on the output node of the circuit, the output was processed by Monte Carlo and the output voltage was collected. The output voltage of each mode was used as the original data set. To visually compare the characteristics of circuit failures in different modes, the simulation results of the above two circuits were selected, and the signal data of the eight failure modes and the normal mode were displayed. Figures 5 and 6 present the results of circuit simulation of different fault modes in the Sallen–Key circuit and CSTV circuit. As can be seen from the figure, there are differences in the output signal of the circuit in different fault modes.

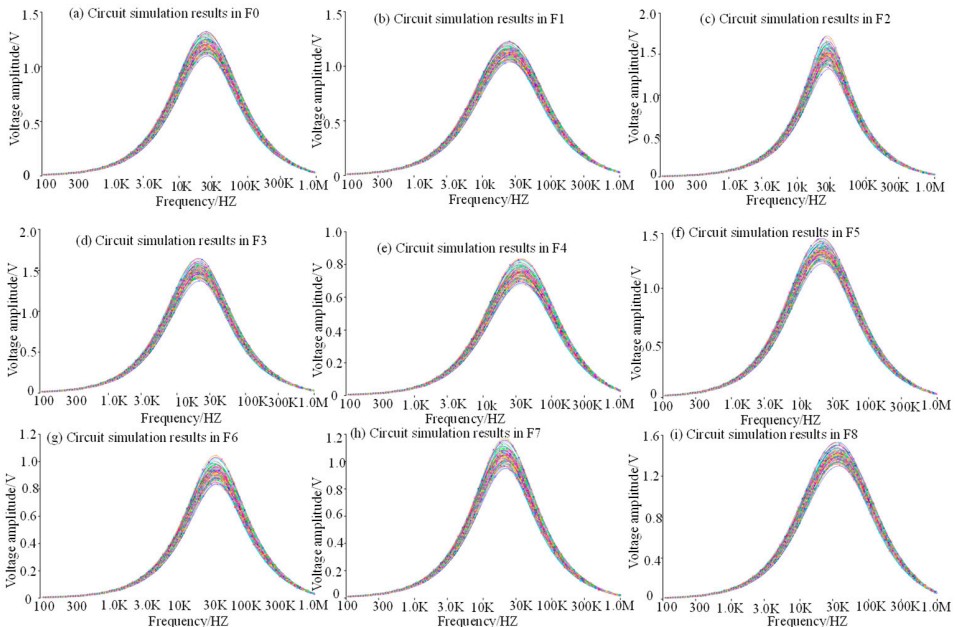

**Figure 5.** The output signal of Sallen-Key band-pass filter circuit.

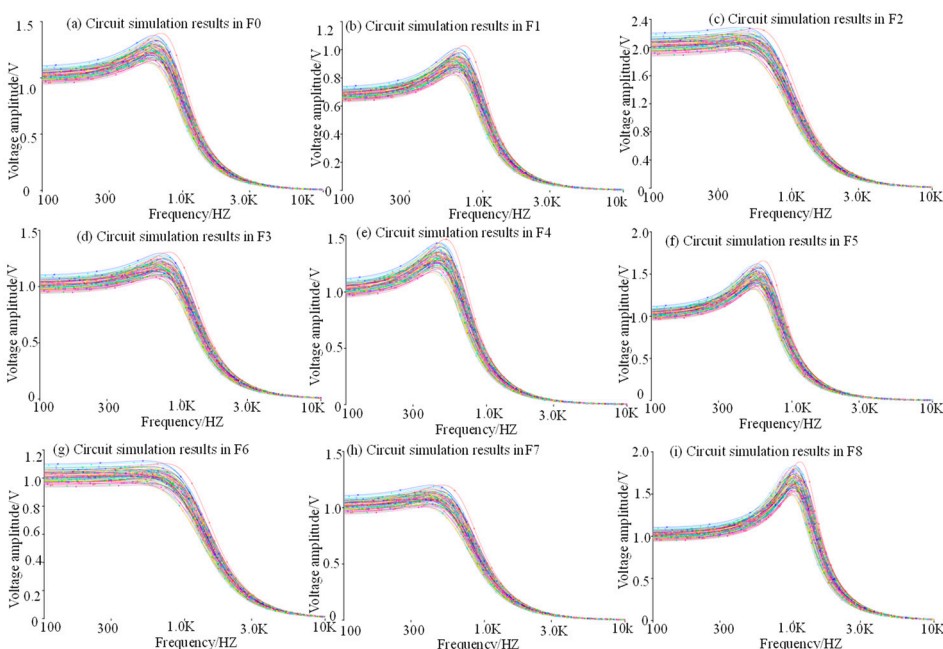

**Figure 6.** The output signal of CSTV filter circuit.

Figures 5 and 6 reflect the states of different faulty circuit signals. Since the analog circuit fault data is set by changing the values of different elements of the circuit. Therefore, the circuit signals corresponding to each fault have a certain similarity. In each subplot in Figures 5 and 6, each image is similar, but each subplot is slightly different due to the different values of the components.

### 4.3.2. Feature Extraction and Dimensionality Reduction of Fault Signals

In this paper, the wavelet packet was used for feature extraction. The feature extraction is representative data extracted from a large amount of data, so the original signal is similar but not identical to the signal after feature extraction. In Figure 7, it can be seen that Figure 7a is similar but not the same as Figure 7d. The main function in the step of extracting features is to obtain highly representative data in the overall data, that is to remove redundant data. In Figure 7, it can be seen that Figure 7c contrasts Figure 7b with the redundant signal data removed.

After the analog circuit is built, the relevant parameters are set. All failure modes are analyzed by 100 MC, the start frequency is set to 100 Hz, and the cutoff frequency is set to 10 kHz. At the output node of the circuit, a voltage probe is used to collect the output voltage. The output voltage is used as the original data set, the wavelet packet is used to extract the features, and the PCA is used to reduce the dimensionality. Finally, the feature-processed analog circuit fault data is obtained, which can be used in the fault diagnosis classifier.

When the fault signal is extracted, the output voltage in each mode is obtained as the original data set, and the features are extracted by wavelet packet, and then the PCA dimension reduction is performed. Finally, the fault data of the analog circuit after feature processing is obtained, which is used in the fault-diagnosis classifier. Among them, the wavelet packet is mainly used to extract the eigenvalues, and the wavelet packet comes from the wavelet, which is a more perfect decomposition method based on the wavelet decomposition. Compared with the wavelet transformation, the wavelet packet transformation can analyze the signal in all directions, can further decompose the wavelet transform without subdividing, and evenly distribute the corresponding two frequency bands according to the same spacing in the high frequency part in the range. This article uses a five-layer wavelet packet, which can effectively decompose the input signal to extract

the characteristic value. Figure 7 shows the changes before and after the signal when the wavelet packet is used in this article to extract the characteristic value.

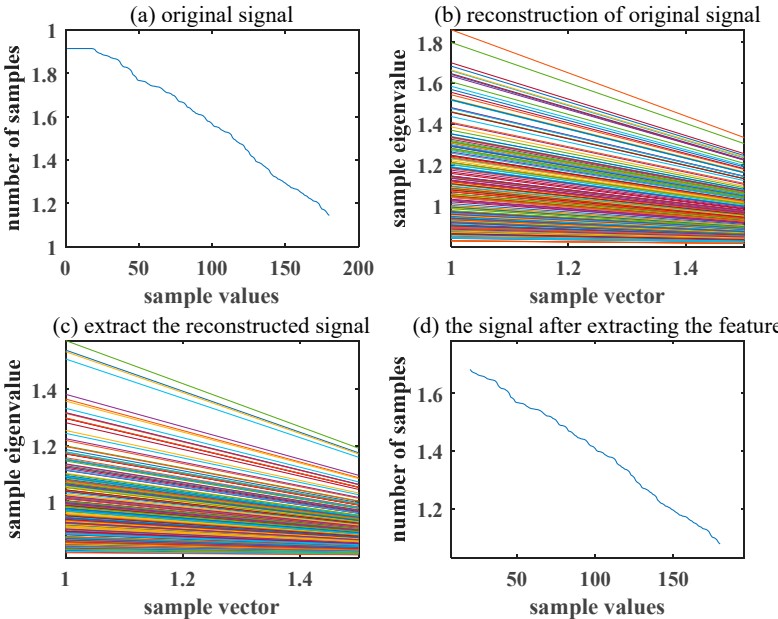

**Figure 7.** Comparison before and after extraction of eigenvalue signal by wavelet packet, where (**a**) original signal; (**b**) reconstruction of original signal; (**c**) extract the reconstructed signal; (**d**) the signal after extracting the feature.

Regarding the PCA technique, the main purpose of this essay is to use PCA to perform dimensionality reduction operations. The number of selected Principal components is five, as shown in Figure 8, which reflects the changes in the data before and after the PCA analysis. When using PCA to reduce the dimensionality, the threshold is set to 85%. When the cumulative contribution rate reaches 85%, the pivot is no longer selected. After all the original variables are processed by dimensionality reduction, the pivot can be obtained. The pivot is the linearity of the original variable combination.

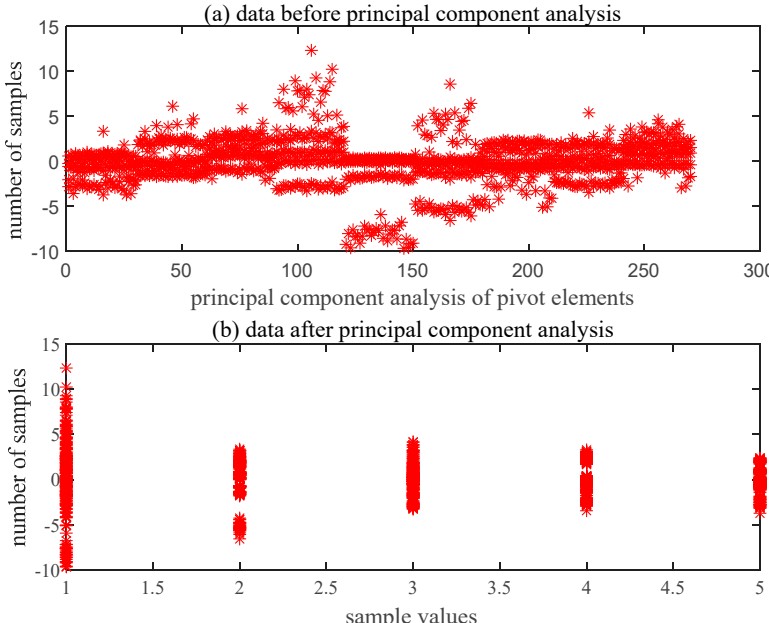

**Figure 8.** Data comparison before and after principal component analysis.

## 5. Algorithm Horizontal and Vertical Comparison Experiment Results

*5.1. SCA Optimization Parameter Comparison*

5.1.1. Comparison of Optimized Parameters under Sallen-Key Band-Pass Filter Circuit

The fault-diagnosis method in this article is based on SVM. There are two important parameters in SVM, namely the penalty factor C and the parameter coefficient g of the kernel function. The two parameters determine the classification performance of the SVM, and a way to quickly and effectively obtain the optimal parameter value becomes the key consideration of the optimization algorithm. This paper compares optimization parameter algorithms such as Grid Search, GA, PSO, ACA, SA and SCA, by comparing the results, and the SCA method is considered to be the best way to optimize parameters. The main operation is to compare Grid Search-SVM, GA-SVM, PSO-SVM, ACA-SVM, SA-SVM and SCA-SVM. The classifiers of different optimization algorithms are used in the fault diagnosis of the Sallen-Key circuit, and there are certain advantages in determining the SCA-optimization parameters. The experimental results are shown in Table 3, which compares the results of fault diagnosis with different optimization algorithms. It can be concluded that the SCA-optimization parameter algorithm can enhance the optimization speed and reduce the optimization time under the premise of meeting the parameter optimization requirements, so that the performance of the entire model not only has the rapidity of classification, but also has the accuracy of classification.

**Table 3.** Sallen-Key bandpass filter circuit optimization parameter algorithm comparison.

| Optimization Parameter Algorithm | Accuracy Rating/% | Elapsed Time/s |
|:---:|:---:|:---:|
| Grid Search | 100 | 62.37 |
| GA | 87.04 | 31.35 |
| PSO | 99.67 | 19.87 |
| ACA | 98.13 | 30.52 |
| SA | 89.65 | 17.54 |
| SCA | 100 | 10.85 |

Fitness curves are introduced to reflect the performance of different optimization algorithms. In Figures 9–14, the fitness curves of different optimization algorithms applied to the SVM parameter optimization of the Sallen-Key circuit fault data are detailed.

The optimization parameter comparison experiment in this article aimed to perfect the parameters of the classification algorithm under the premise of SVM, in which no changes are made to the various modules of the fault classification, and only the optimization-parameter module is compared to ensure that the external conditions are consistent. The experimental performance is compared below, and the comparison results are shown in Table 3.

The experimental results can be deduced by comparing optimization-parameter algorithms such as Grid Search, GA, PSO, ACA and SA. The performance of SCA-optimization parameters is superior than other algorithms in terms of accuracy and classification speed.

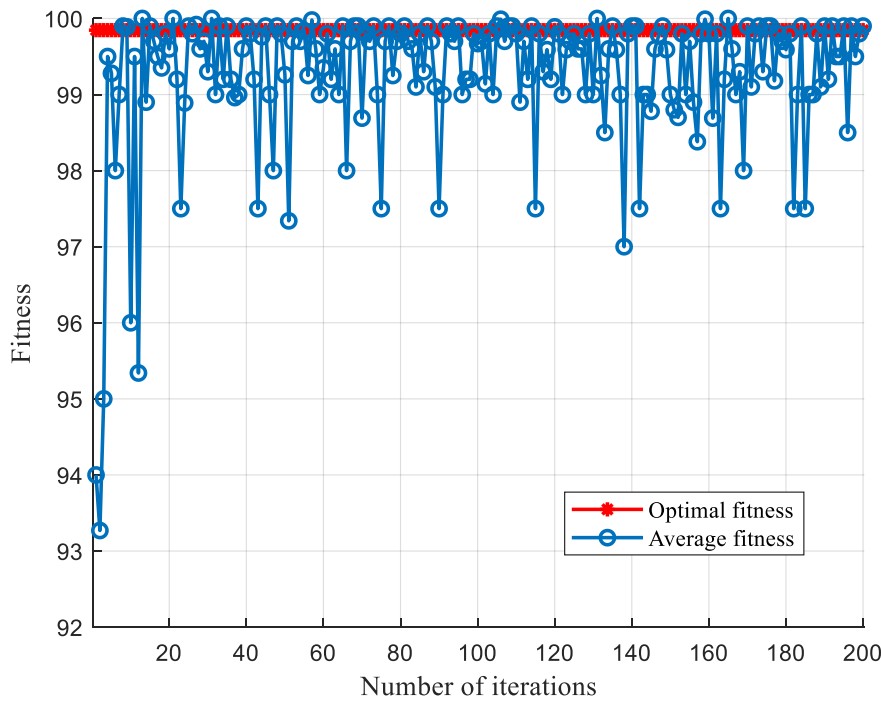

**Figure 9.** Grid Search-SVM fitness curve from Sallen-Key band-pass filter.

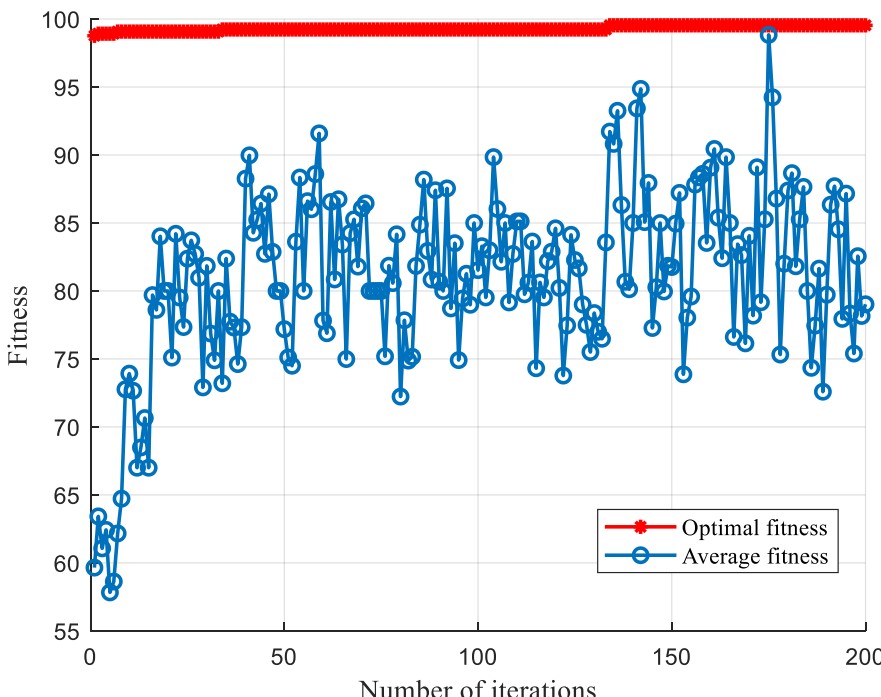

**Figure 10.** GA-SVM fitness curve from Sallen-Key band-pass filter.

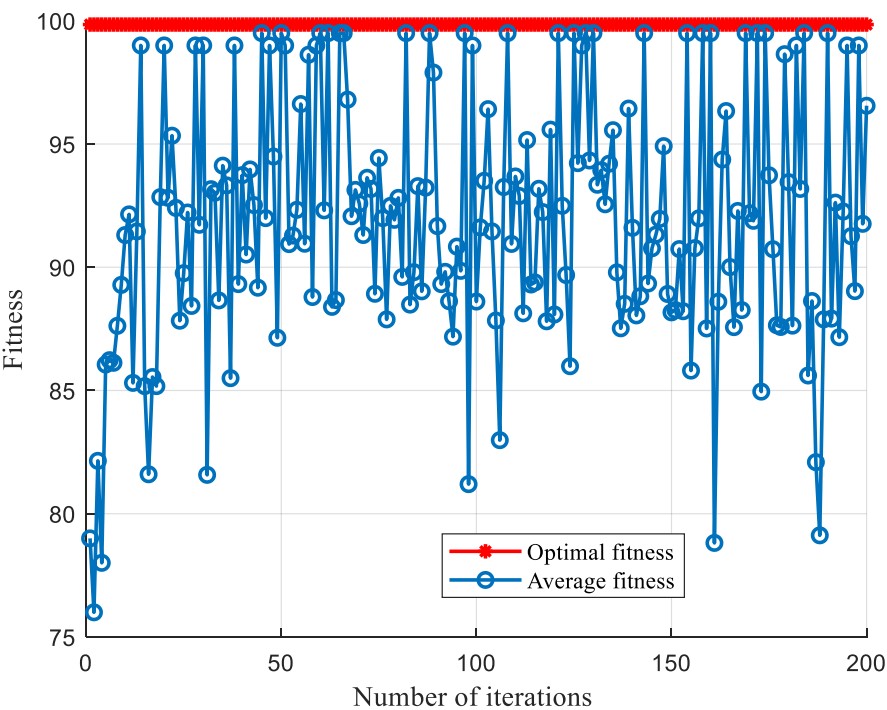

**Figure 11.** PSO-SVM fitness curve from Sallen-Key band-pass filter.

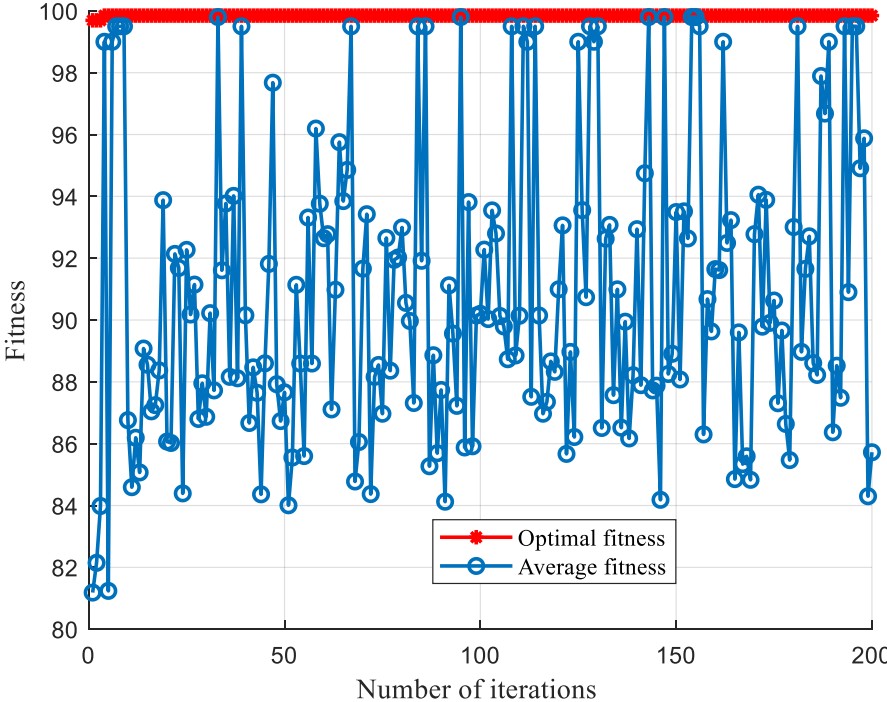

**Figure 12.** ACA-SVM fitness curve from Sallen-Key band-pass filter.

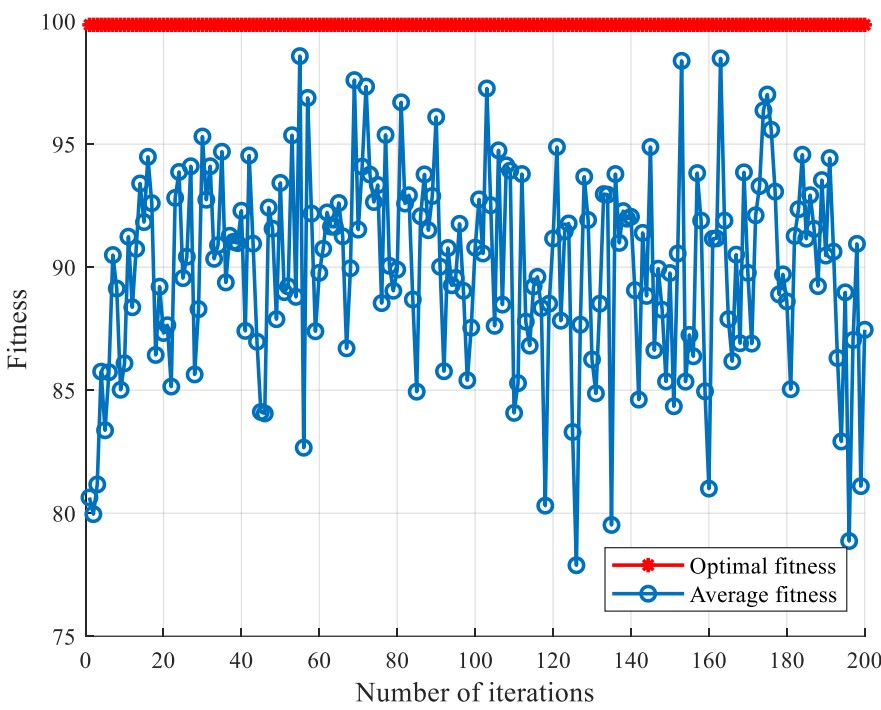

**Figure 13.** SA-SVM fitness curve from Sallen-Key band-pass filter.

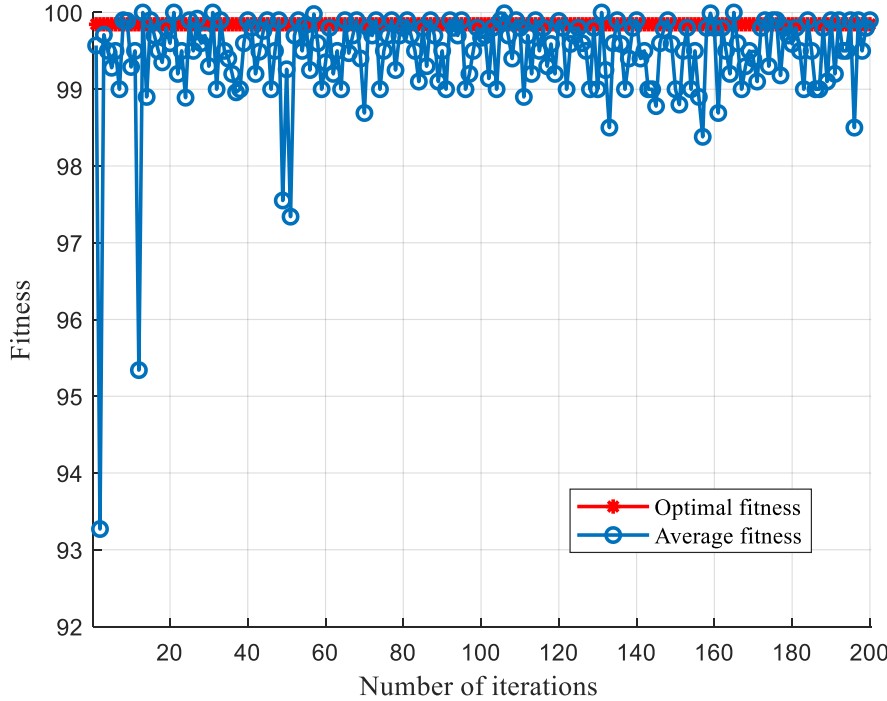

**Figure 14.** SCA-SVM fitness curve from Sallen-Key band-pass filter.

### 5.1.2. Comparison of Optimized Parameters under CSTV Filter Circuit

In the method of comparing and optimizing parameters, the common Sallen-Key filter is used for comparison experiments. To verify the generality of the comparison method conclusions, the multi-stage filter, which is a CSTV circuit, is used for verification. Grid Search-SVM, GA-SVM, PSO-SVM, ACA-SVM, SA-SVM and SCA-SVM are compared, and the fault classifier is used with the CSTV filter-circuit data to ensure that the fault-diagnosis classifier has universal applicability. In Table 4, comparison results of optimized parameter algorithms by the CSTV filter circuit are presented.

**Table 4.** CSTV filter circuit optimization parameter algorithm comparison.

| Optimization Parameter Algorithm | Accuracy Rating/% | Elapsed Time/s |
| :---: | :---: | :---: |
| Grid Search | 99.85 | 73.07 |
| GA | 81.54 | 34.08 |
| PSO | 97.08 | 27.15 |
| ACA | 95.38 | 39.87 |
| SA | 83.66 | 26.31 |
| SCA | 99.89 | 18.49 |

Fitness curves are introduced to reflect the performance of different optimization algorithms. As shown in Figures 15–20, the fitness curves of different optimization algorithms applied to the SVM parameter optimization of the Sallen–Key circuit fault data are shown in detail. When comparing the algorithms for optimizing parameters, the number of iteration steps is uniformly set to 200.

Finally, in order to draw a more general conclusion, the CSTV filter circuit is used as verification. It can be concluded that the performance of the SCA-optimization parameters has certain advantages, when considering the classification accuracy and classification speed. SCA not only meets the requirements of parameter optimization, but also improves the optimization speed and reduces the optimization time, so that the performance of the entire algorithm has both the rapidity of classification and the accuracy of classification.

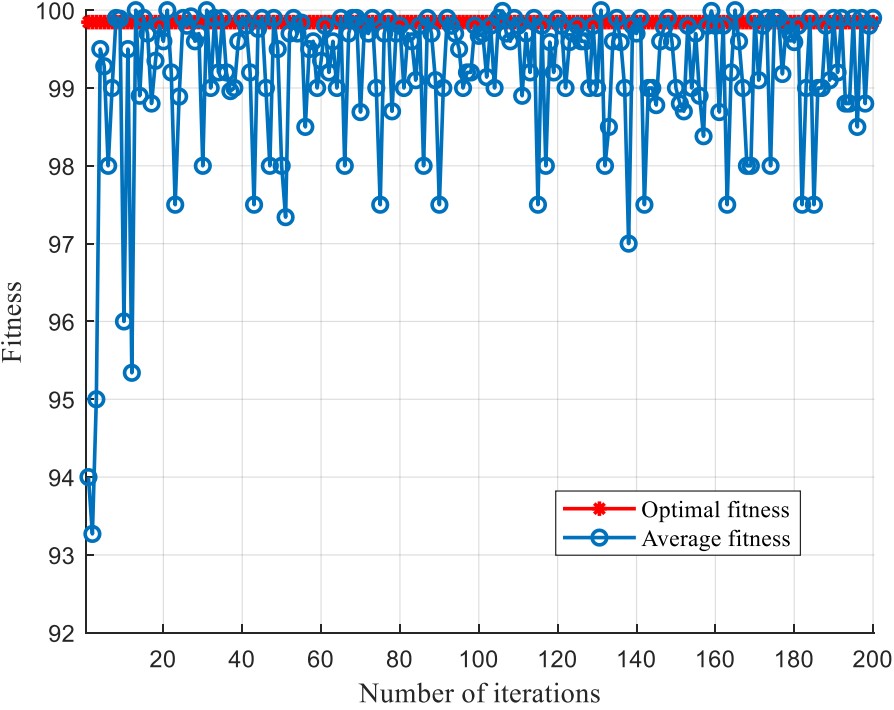

**Figure 15.** Grid Search-SVM fitness curve from CSTV filter.

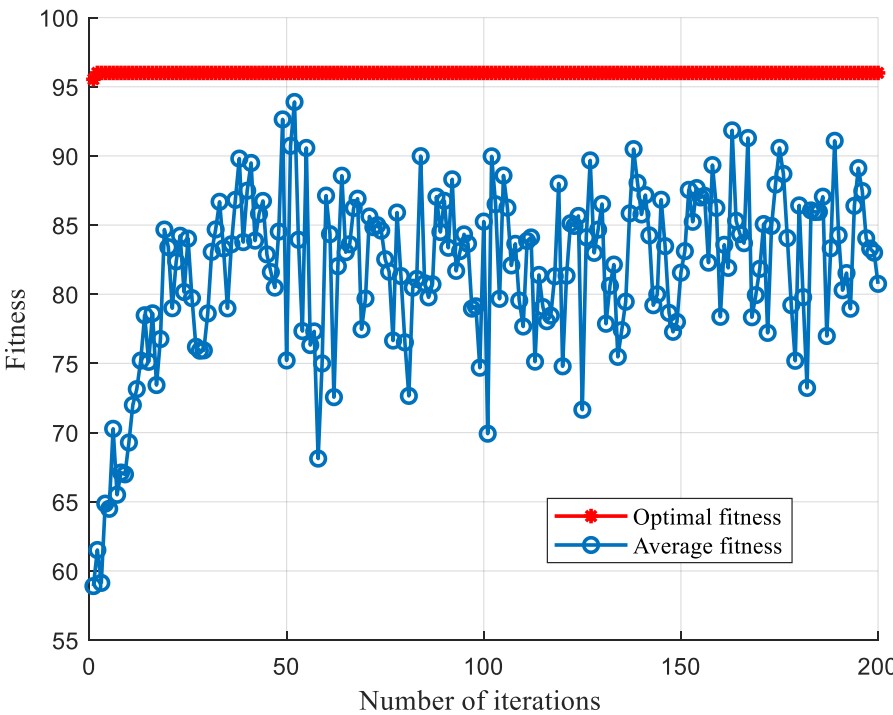

**Figure 16.** GA-SVM fitness curve from CSTV filter.

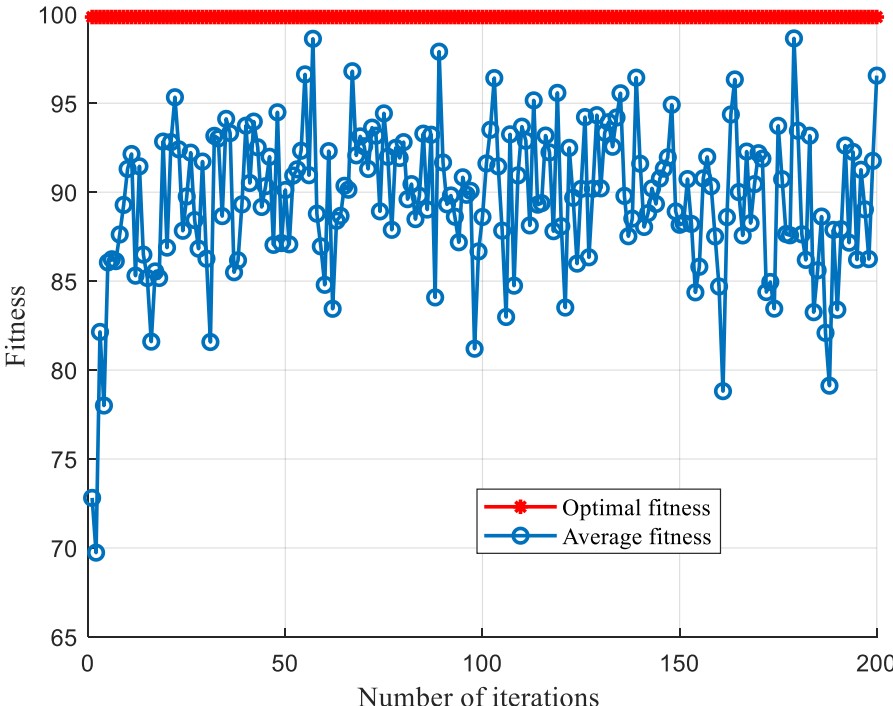

**Figure 17.** PSO-SVM fitness curve from CSTV filter.

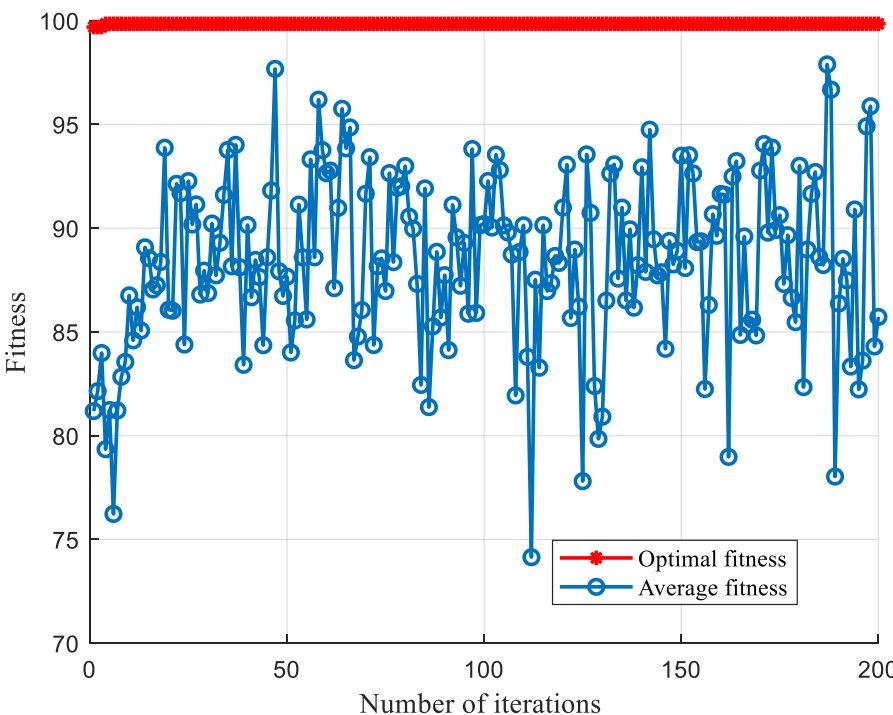

**Figure 18.** ACA-SVM fitness curve from CSTV filter.

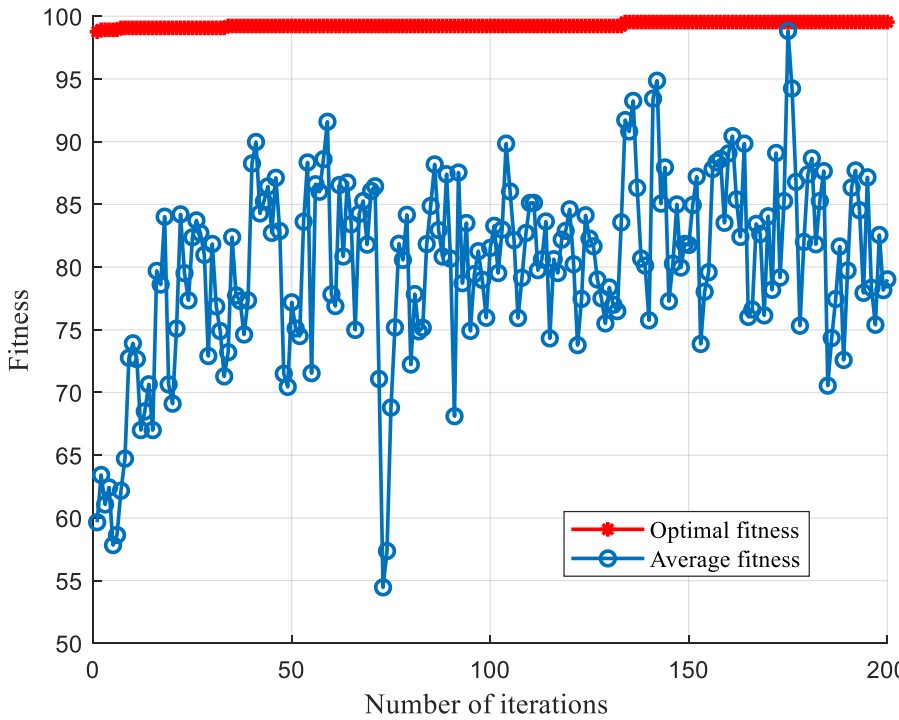

**Figure 19.** SA-SVM fitness curve from CSTV filter.

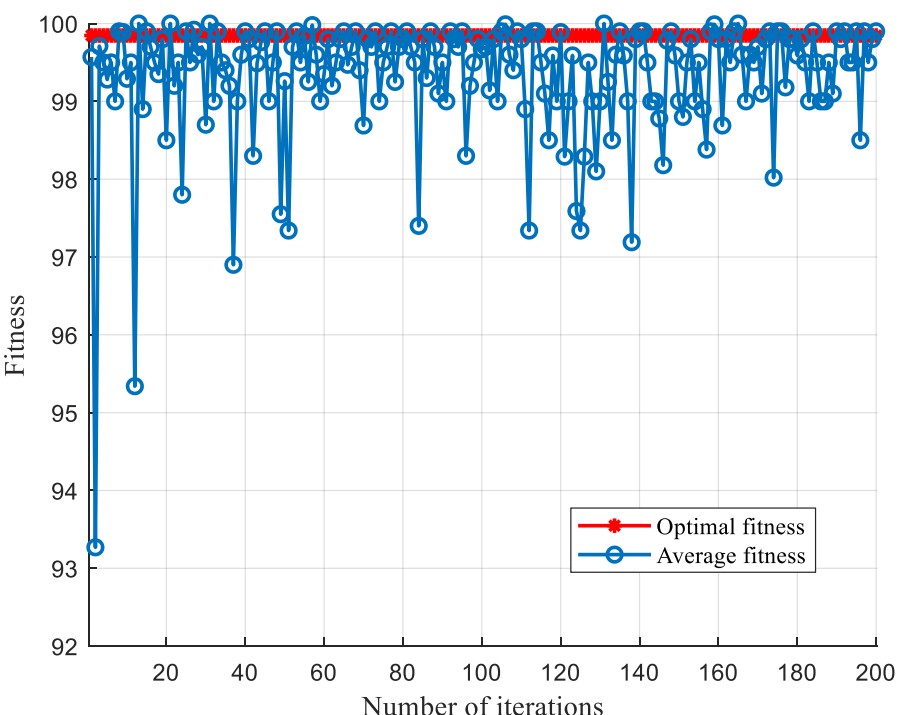

**Figure 20.** SCA-SVM fitness curve from CSTV filter.

### 5.2. Comparison of SCA-SVM Classification Algorithms

5.2.1. Comparison of Classification Algorithms under Sallen-Key Bandpass Filter Circuit

According to the fault data of the Sallen-Key circuit, different classification algorithms are used, such as BP, SOM, ELM, decision tree, random forest and SCA-SVM. Through comparative experiments, it is concluded that the SCA-SVM classification algorithm is superior to other classification algorithms in fault diagnosis. As shown in Figures 21–26, the classification effect of the Sallen-Key circuit fault data by different classification algorithms is shown in detail.

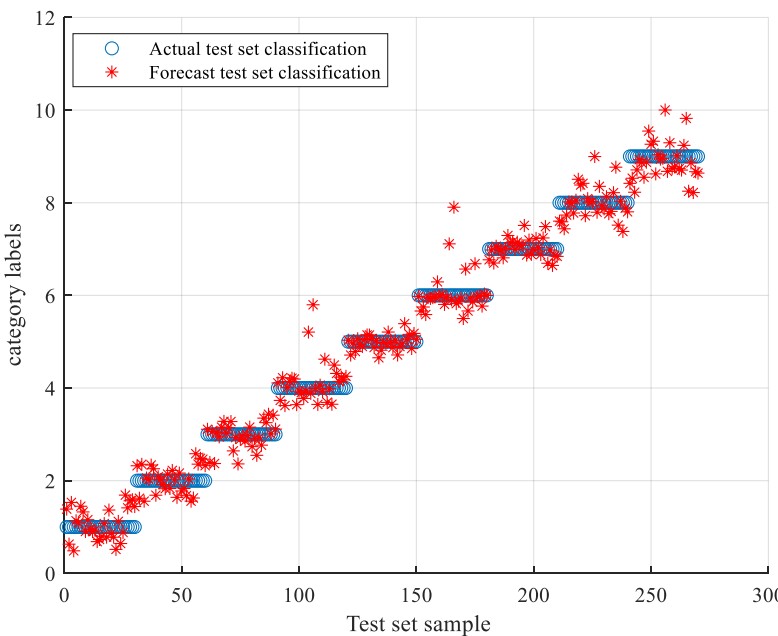

**Figure 21.** Classification effect of BP algorithm from Sallen-Key bandpass filter.

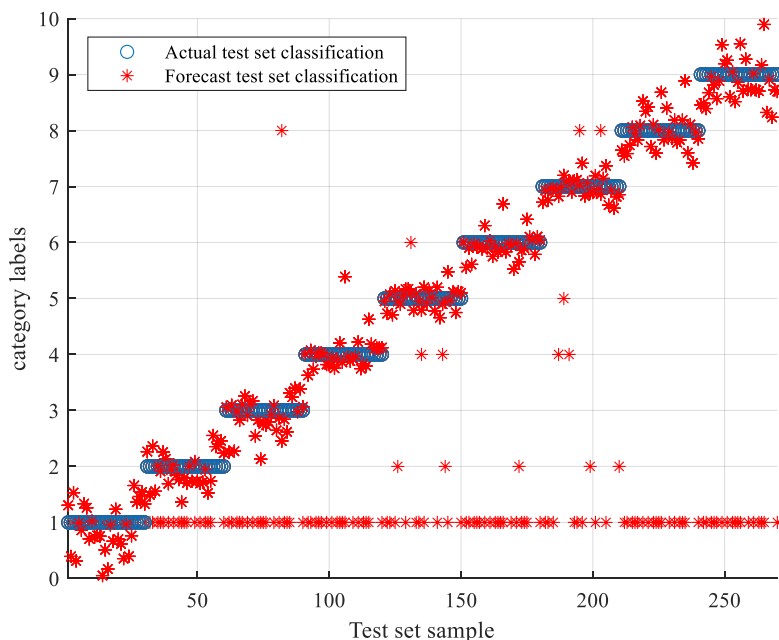

**Figure 22.** Classification effect of SOM algorithm from Sallen-Key bandpass filter.

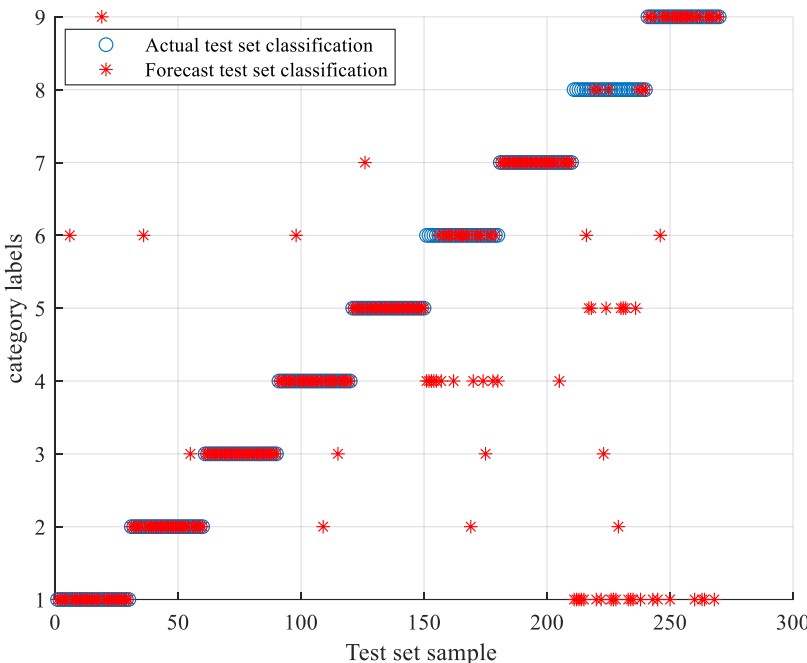

**Figure 23.** Classification effect of ELM algorithm from Sallen-Key bandpass filter.

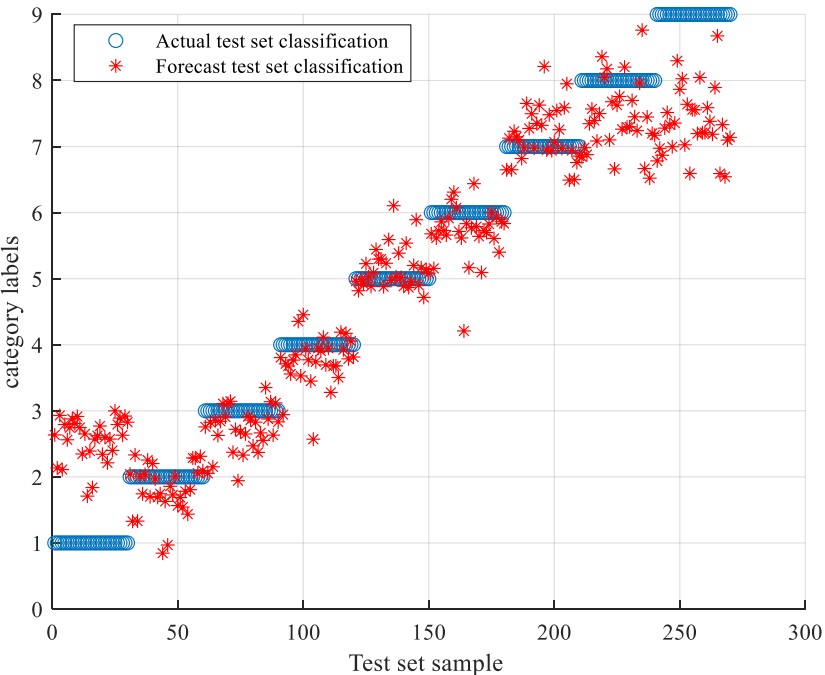

**Figure 24.** Classification effect of decision tree algorithm from Sallen-Key bandpass filter.

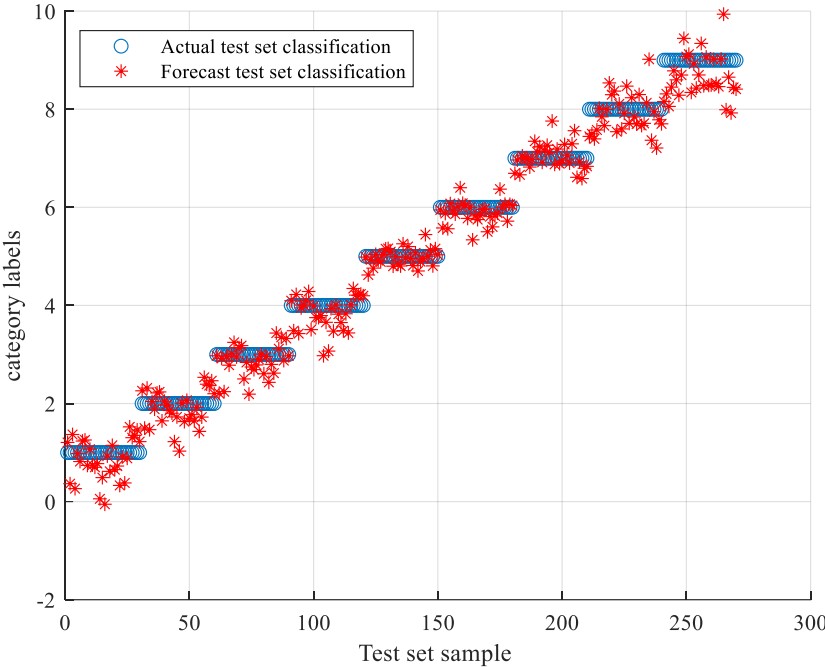

**Figure 25.** Classification effect of random forest algorithm from Sallen-Key bandpass filter.

It is worth mentioning that when comparing the classification algorithms, the neural networks [43] used for comparison include BP, RBF, GRNN, PNN, competitive neural network, and SOM. The characteristics of the experimental data are combined. The BP and the SOM have the best comprehensive classification performance. After comprehensive consideration, it was decided to use the BP neural network and the SOM neural network as representative of the neural network-classification algorithm for comparison.

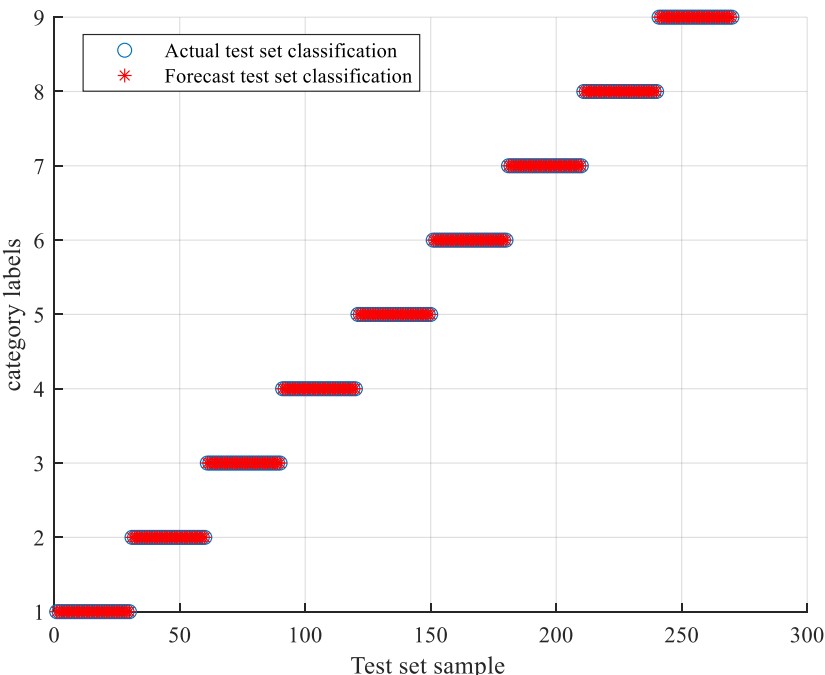

**Figure 26.** Classification effect of SCA-SVM algorithm from Sallen-Key bandpass filter.

In terms of the classification effect, the accuracy of the SCA-SVM classification algorithm is superior to other algorithms. Table 5 is obtained by digitizing the experimental statements. As presented in Table 5, by comparing other classification algorithms such as BP, SOM, ELM, decision trees, random forests, etc., the conclusion is obtained. It draws a conclusion that when considering classification accuracy, the performance of SCA-SVM is better than that other algorithms. SCA is very effective for optimizing SVM parameters fo4 fault diagnosis, and the classification effect of the Sallen–Key circuit fault data can reach 100%.

**Table 5.** Comparison of Sallen-Key bandpass filter circuit classification algorithms.

| Classification Algorithm | Accuracy Rating/% | Elapsed Time/s |
| :---: | :---: | :---: |
| BP | 99.25 | 31.57 |
| SOM | 82.76 | 7.20 |
| ELM | 94.70 | 1.43 |
| Decision Tree | 93.07 | 4.26 |
| Random Forest | 97.88 | 9.75 |
| SCA-SVM | 100 | 10.85 |

### 5.2.2. Comparison of Classification Algorithms under CSTV Filter Circuit

In the comparison classification algorithm, the common second-order filter, which is the Sallen–Key circuit, is used for comparison experiments. To verify the versatility of the conclusions of the comparison method, a multi-stage filter, which is a CSTV filter circuit, is used for verification. Since the CSTV filter circuit is more complex and there are more aspects to consider in the fault diagnosis, the data of the CSTV filter circuit is added for a better analysis. So, for the CSTV filter circuit, a total of 1800 samples were collected, of which 900 samples constitute the training data set, and the remaining 900 samples constitute the test data set. In Figures 27–32, the classification effect of CSTV filter circuit fault data with different classification algorithms is shown in detail.

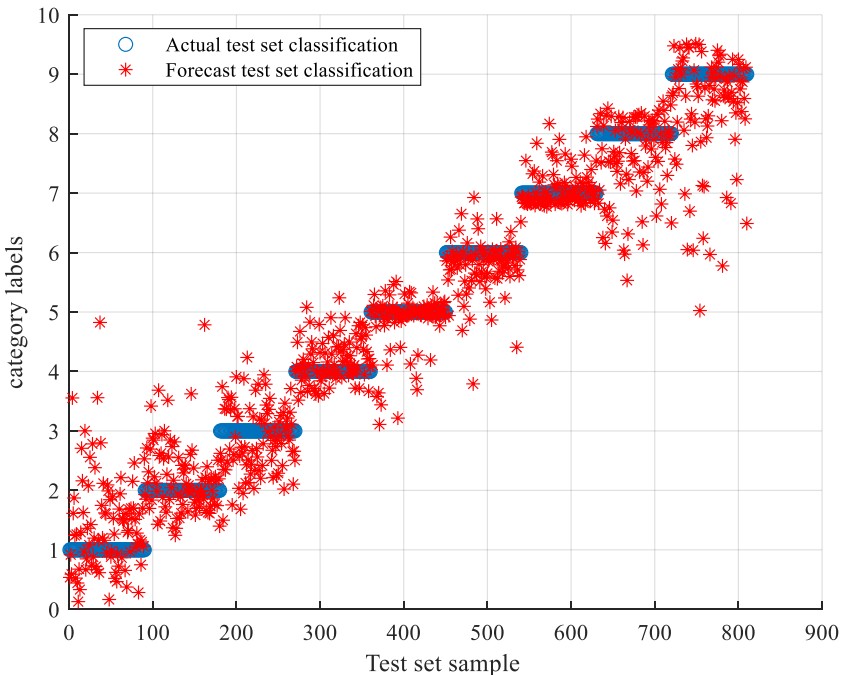

**Figure 27.** Classification effect of BP algorithm from CSTV filter.

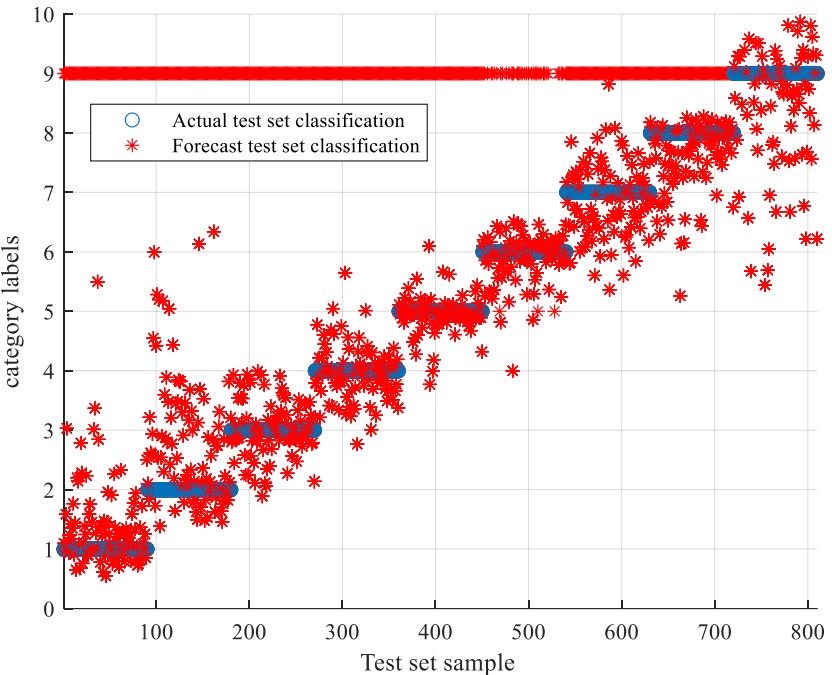

**Figure 28.** Classification effect of SOM algorithm from CSTV filter.

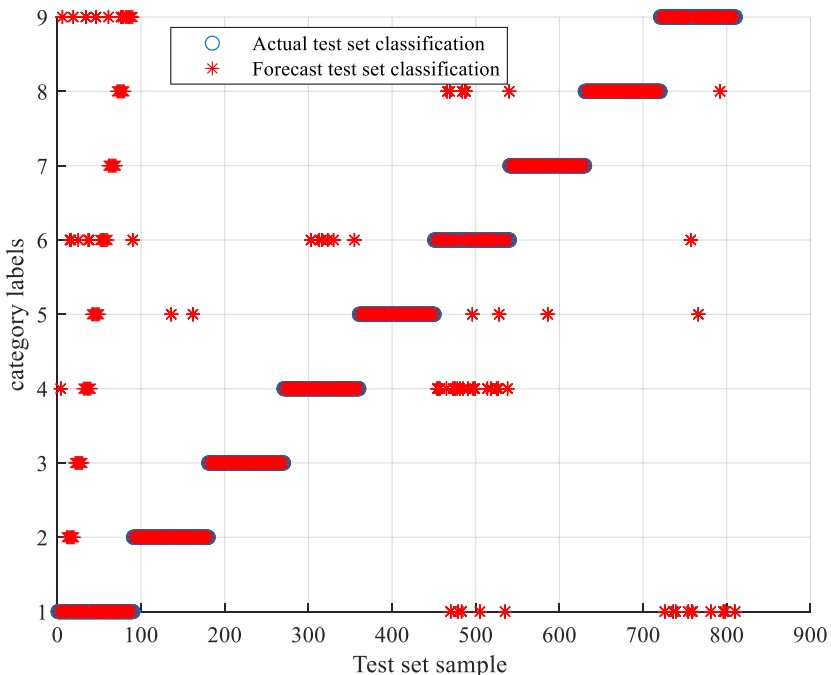

**Figure 29.** Classification effect of ELM algorithm from CSTV filter.

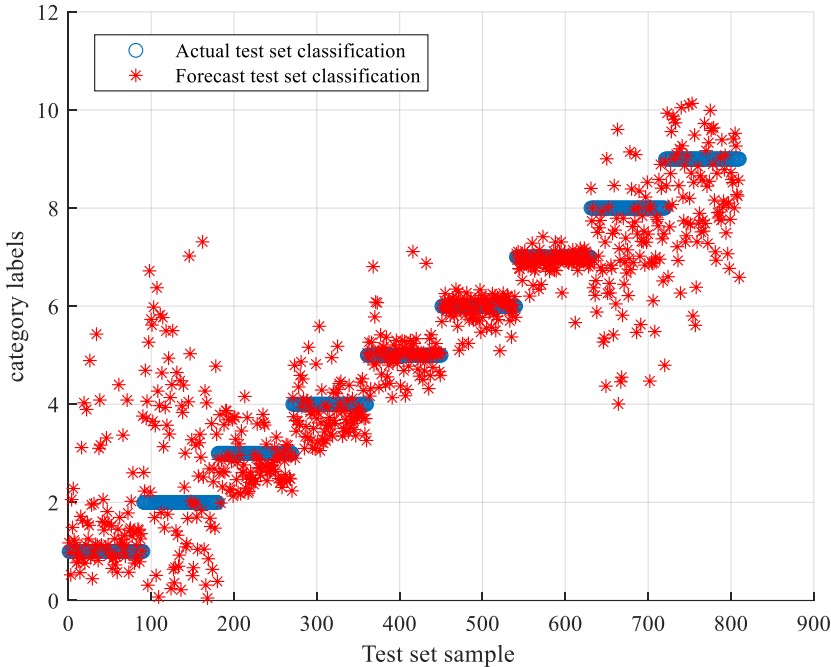

**Figure 30.** Classification effect of decision tree algorithm from CSTV filter.

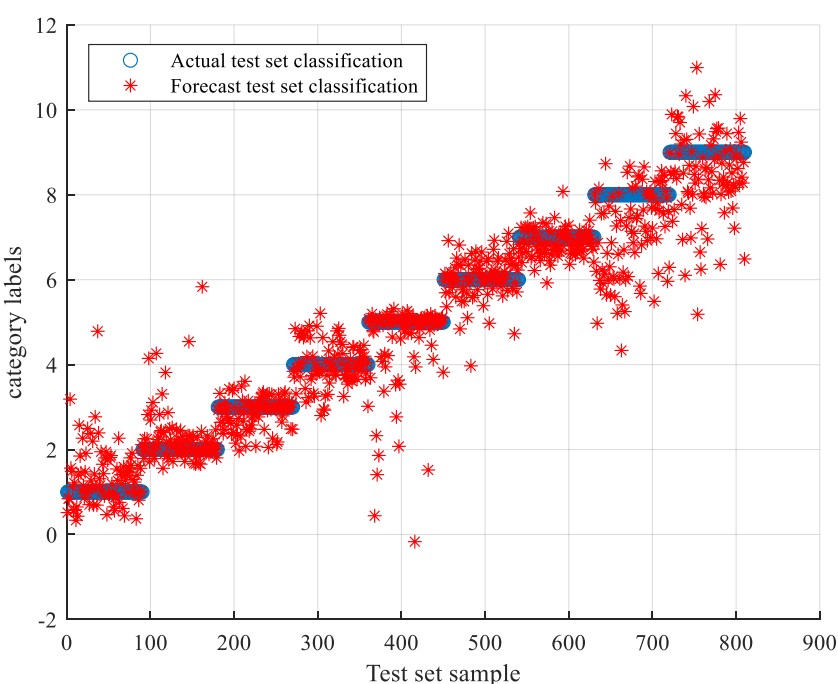

**Figure 31.** Classification effect of random forest algorithm from CSTV filter.

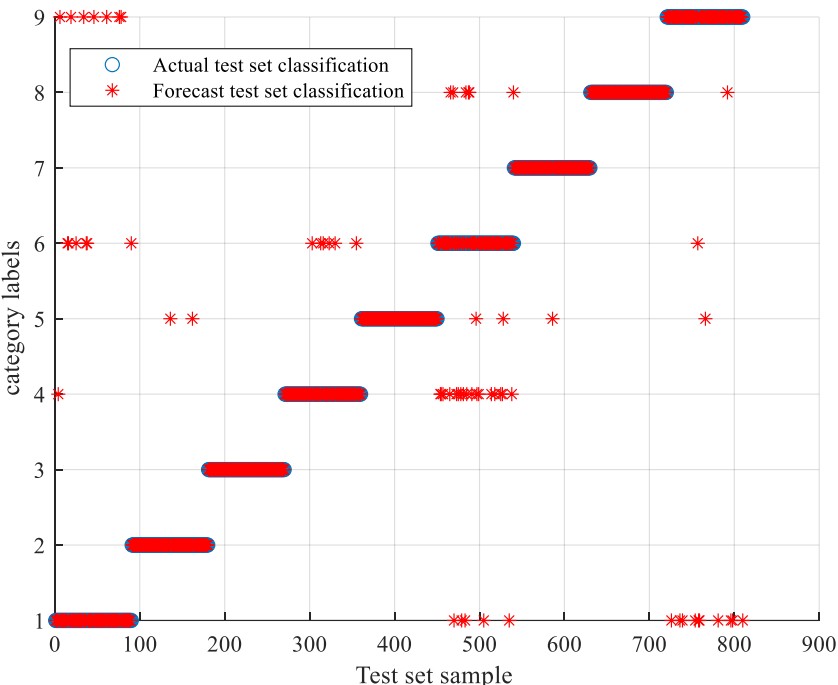

**Figure 32.** Classification effect of SCA-SVM algorithm from CSTV filter.

It can be seen from Table 6 that by comparing other classification algorithms such as BP, SOM, ELM, decision trees, random forests, etc., when considering accuracy, the performance of SCA-SVM is better than other algorithms. SCA is very effective for optimizing SVM parameters for fault diagnosis, and the classification effect of the CSTV filter circuit fault data can reach 99.89%.

**Table 6.** Comparison of CSTV filter circuit classification algorithms.

| Classification Algorithm | Accuracy Rating/% | Elapsed Time/s |
|---|---|---|
| BP | 96.89 | 40.69 |
| SOM | 74.15 | 17.49 |
| ELM | 91.85 | 9.12 |
| Decision Tree | 89.45 | 19.99 |
| Random Forest | 95.12 | 10.90 |
| SCA-SVM | 99.89 | 18.49 |

In order to draw a more general comparison algorithm conclusion, the multi-stage filter, that is, the CSTV filter circuit, is used as a verification object and a more general conclusion can be drawn. When the target is analog circuit-fault diagnosis, SCA-SVM is considered to have certain advantages at the level of classification accuracy.

*5.3. TLSCA-SVM Comparative Test Results*

The aforementioned SCA-SVM fault classifier can be effectively applied to fault diagnosis, but when there are too few fault samples or only normal samples, there will be a problem of inaccurate fault diagnosis. Therefore, this paper introduces the TLSCA-SVM algorithm. An auxiliary condition, that is, an error penalty term, is added to the objective function of the SCA-SVM classifier to construct a new fault-diagnosis model so that the fault diagnosis satisfies the ability to effectively classify faults when the fault samples are not complete. It combines the advantages of the SCA-SVM classifier in fault diagnosis with high accuracy, fast diagnosis speed and good stability.

According to the type of training samples, transfer learning can be divided into zero-shot learning and few-shot learning. This paper changes the proportion of faulty data in the training set by changing the database to reduce the proportion of faulty data in the training set. Faulty data and normal data are kept in the test set to perform transfer learning. It is worth mentioning here that the limit of few-shot learning in transfer learning is zero-shot learning. Zero-shot learning can be achieved under an extremely idealized model, but zero-shot learning is unrealizable in real data processing, so the transfer learning of this algorithm is embodied in the fault classification of few-shot learning. This paper compares the SCA-SVM classifier used in traditional machine learning with the TLSCA-SVM classifier, based on the transfer-learning theory by constantly changing the proportion of the failure samples in the training set. Figure 33 is obtained. There is a point to note here. Since the transfer learning ability of the comparison algorithm requires the use of a lot of data and a relatively simple circuit form, in the comparison experiment, a simple Sallen-Key band-pass filter circuit with more circuit data was used.

The conclusion can be drawn from Figure 33 that when the training sample is relatively small, the traditional SCA-SVM classification algorithm cannot effectively perform fault diagnosis. The classifier trained by the non-transfer learning model is not effective in the fault diagnosis of few-shot learning. Classifiers with transfer ability have certain advantages in few-shot learning. As the proportion of fault samples increases, the effect of transfer learning becomes weaker. When the proportion of fault samples reaches 50%, the effect of transfer learning and non-transfer-learning fault diagnosis is the same. When the proportion of faulty samples does not reach 50%, the classifier with transfer learning ability shows better classification performance in fault diagnosis. This article classifies the performance of the TLSCA-SVM classification algorithm in fault diagnosis as better than SCA-SVM fault samples when the proportion of faults does not reach 50%.

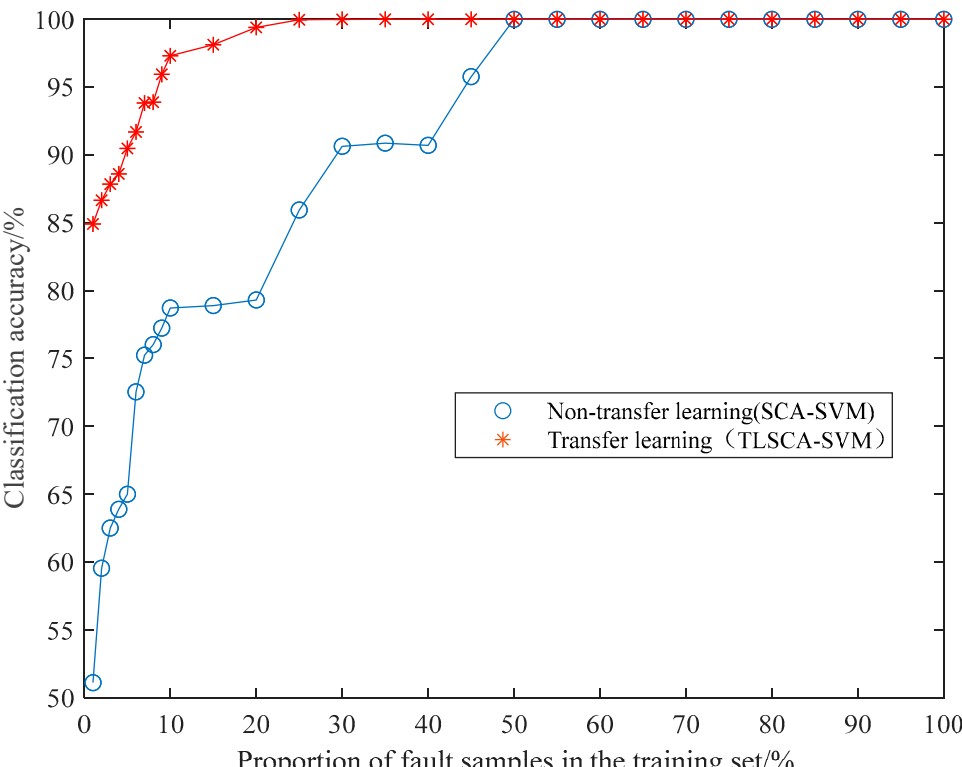

**Figure 33.** Comparison of transfer learning and non-transfer learning.

## 6. Conclusions

This article uses SCA to optimize SVM parameters, and proposes a classification-diagnosis method using SCA-SVM. Finally, it is proposed that the TLSCA-SVM classification algorithm can be used in the case of fewer fault samples in the training set. The data of the Sallen–Key circuit is used as experimental data, the data of the CSTV circuit is used to verify the versatility of the method, and a more general conclusion is drawn as follows:

(a)  By comparing optimization parameter algorithms such as Grid Search, GA, PSO, ACA, SA, etc., the SCA is proposed. The SCA optimization-parameter algorithm can improve the optimization speed and shorten the optimization time on the premise of meeting the parameter-optimization requirements. The entire algorithm has both high classification accuracy and fast classification performance.

(b)  The SCA-SVM fault-classification method has superior performance in the fault diagnosis. A complex CSTV circuit is used to verify the versatility of the method. Several comparison experiments show that the method is not only superior, but also universal in performance to other algorithms. Different classification algorithms are used, such as BP, SOM, ELM, decision tree, random forest and SCA-SVM to compare. It can be concluded that the accuracy of the SCA-SVM classification algorithm is superior to other comparison algorithms in terms of the classification effect.

(c)  With regard to the problems of most optimization algorithms, this paper reasonably avoids them. The classification algorithm of this paper is analyzed. This paper uses the SVM classifier as the main body for fault diagnosis. The SVM classifier itself has a good classification effect, and the difference between the important parameter's penalty factor C and kernel parameter will affect the classification effect of the SVM. The objective of the SCA algorithm is to obtain appropriate parameters. The search method is randomly determined each time an optimal solution is found. That is to say, in the next search, both the local search and the global search are random, i.e., the probability is the same. Each time the optimal solution is approximated, the approximation method is randomly determined. Such an optimization method can avoid local optimal solutions and shorten the optimization time. The classifier formed

after finding suitable parameters has a good classification effect in fault classification, and the classification efficiency improved.

(d) Various optimization algorithms were compared, such as Gray Wolf Optimization (GWO), Gravitational Search Algorithm (GSA), competitive swarm optimizer (CSO), etc. The advantages and disadvantages of different optimization algorithms were discovered. Most of the shortcomings focus on non-global search. When the optimization algorithm performs a non-global search, local optimal solutions may appear. After continuous exploration, some algorithms were optimized by combining the characteristics of multiple algorithms. For example, the GWO was combined with SCA to optimize parameters. These issues deserve to be studied in future.

(e) When the training data is deficient, the TLSCA-SVM classification algorithm can effectively diagnose the fault. Because the TL-SCASV algorithm adds an auxiliary condition to the objective function of the SCA-SVM classifier, that is, an error penalty term to construct a new fault diagnosis model, the fault diagnosis is satisfactory. When the fault samples are not complete, it can still effectively classify the faults. It combines the advantages of the SCA-SVM classifier with high accuracy, fast diagnosis speed and good stability in fault diagnosis. The algorithm not only achieves high fault-diagnosis accuracy, but can also operate effectively in the case of a lack of fault samples, and can effectively perform fault classification in multiple circuits.

To sum up, the TLSCA-SVM classifier was constructed. When there are more fault samples, the fault classification effect of the classifier was found to be better by a cross-sectional comparison. The classifier was also effective in diagnosis when there were fewer fault samples. It has certain versatility in fault-data diagnosis and broad application prospects.

**Author Contributions:** Conceptualization, A.Z. and D.Y.; methodology, A.Z. and D.Y.; software, D.Y.; validation, A.Z., D.Y. and Z.Z.; formal analysis, D.Y.; investigation, D.Y.; resources, A.Z.; data curation, D.Y.; writing—original draft preparation, D.Y.; writing—review and editing, A.Z. and D.Y.; visualization, D.Y.; supervision, A.Z. and Z.Z.; project administration, A.Z.; funding acquisition, A.Z. All authors have read and agreed to the published version of the manuscript.

**Funding:** This works is partly supported by the Natural Science Foundation of Liaoning, China under Grant 2019MS008, Education Committee Project of Liaoning, China under Grant LJKZ1011 and LJ2019003.

**Data Availability Statement:** Not applicable.

**Conflicts of Interest:** The authors declare no conflict of interest.

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
