# Peer review of "TLSCA-SVM Fault Diagnosis Optimization Method Based on Transfer Learning"

_processes, doi:10.3390/pr10020362_

Round 1

Reviewer 1 Report

This paper proposes a TLSCA-SVM fault diagnosis optimization method based on transfer learning

Remarks and questions

- No novelty in the section 2 (2 pages). The authors have to cite the reference where the DOI is 10.1109/ICCCAS.2018.8768963 and reduce this section

- The section 3 “Fault Diagnosis model of TLSCA-SVM algorithm”, the core of the contribution, needs more explanations, interpretations and details: Why does the equation (10) take this form? How does the equation (13) improve the decision making?

-  Section 4 : The proposed approach is applied on two circuits, Sallen-Ky and CSTV, why?

Figures 5 et 6: why is there a band of curves in each subplot? What does it mean?

Figure 7 is not clear.

I understood that the eigenvalues are considered as extracted features. What is the correlation between these features and considered faults?

Is it realistic to consider a unique fault at a given time?

What is the efficiency of the proposed approach in the case of simultaneous faults?

For the considered circuits, the performances of the proposed algorithm are not discussed (false alarm probability, detection probability). Can you give some indicators about this?  

I prefer that the authors take one system and answer my questions above.

Author Response

Response to Reviewer 1 Comments

We would like to thank to the reviewer for the time and effort dedicated on the review of our paper and the very helpful comments and suggestions. According to the comments of the Associate Editor, we have made corresponding revisions to improve the quality of the manuscript. Replies to each individual question are provided as following:

Point 1: The proposed approach is applied on two circuits, Sallen-Ky and CSTV, why?

Response 1: “ Sallen-Key circuits and CSTV circuits are typical circuits that are often used to analog circuit fault diagnosis. In the simulation experiment, considering the selectivity of the simulated circuit data, for more rigorous consideration, the faults of the Sallen-Key circuit and the CSTV circuit are set according to the literatures [38-42].”

“At the same time, the Sallen-Key circuit is used because the circuit is relatively simple as a second-order circuit and public data sets can be used. So, the Sallen-Key circuit is applied to verify the validity of the algorithm. The CSTV circuit is a fourth-order circuit, which is more complex than the Sallen-Key circuit. The application of this circuit can show that the algorithm itself has universal applicability.”

In response to this comment, the discussion on the application of the Sallen-Key circuit and CSTV circuit in this paper is added to the article. For the detailed revisions, please see the second paragraph of section 4 on page 9 for details in the revision.

Point 2: Figures 5 et 6: why is there a band of curves in each subplot? What does it mean?

Response 2: “Figures 5 et 6 reflect the states of different faulty circuit signals. Since the analog circuit fault data is set by changing the values of different elements of the circuit. Therefore, the circuit signals corresponding to each fault have a certain similarity. From each subplot in Figures 5 and 6, each image is similar, but each subplot is slightly different due to the different values of the components.” 

In response to this comment, for the detailed revisions, please see the last paragraph of Section 4.3.1 on page 13 for details in the revision

Point 3: Figure 7 is not clear. I understood that the eigenvalues are considered as extracted features. What is the correlation between these features and considered faults?

Response 3: “In this paper, the wavelet packet is used for feature extraction. The feature extraction is representative data extracted from a large amount of data, so the original signal is similar but not identical to the signal after feature extraction. In Figure 7, it can be seen that subgraph (a) is similar but not the same as subgraph (d). The main function in the step of extracting features is to obtain highly representative data in the overall data. The fault data is set by changing the parameters of different components. So, there is no specific correlation between a single extracted feature and a fault.”

In response to this comment, for the detailed revisions, please see the first paragraph of Section 4.3.2 on page 13 for details in the revision.

Point 4: Is it realistic to consider a unique fault at a given time? What is the efficiency of the proposed approach in the case of simultaneous faults?

Response 4: “In this article, the problem of single fault diagnosis in analog circuits is considered. Single fault refers to changing the parameters of only one component in the circuit while the parameters of other components in the circuit remain unchanged. The situation where multiple component parameters are changed at the same time is referred to as a multiple fault diagnosis problem. The data processing method of multi-fault diagnosis is similar to that of single-fault diagnosis. Multiple faults are changing the parameters of only two or more components while the parameters of other circuit components remain unchanged. Single fault is the basis of fault diagnosis in analog circuit fault diagnosis. In the actual analog circuit fault, the occurrence probability of single fault is more than 80%, and the occurrence probability of multiple faults is relatively low. And multi-fault diagnosis can be regarded as multiple single-fault problems occurring at the same time. That is, a multi-fault problem can be decomposed into multiple single-fault problems. Single-fault diagnosis and multi-fault diagnosis have many similarities in analog circuit fault diagnosis and multi-fault problems with less probability can be decomposed into single-fault for processing. Considering all aspects, this paper does not analyze the multi-fault diagnosis. Inspired by the problem, single-fault diagnosis and multi-fault diagnosis have been considered and this noteworthy discussion has been added to the article, so that readers have a clearer understanding.”

In response to this comment, the detail analyses for those results are carefully given in Section 3.3. For the details, please refer to Section 3.3 on pages 9 in this revised manuscript.

Point 5: For the considered circuits, the performances of the proposed algorithm are not discussed (false alarm probability, detection probability). Can you give some indicators about this?  

Response 5: In the comparison of algorithm performance, a fitness curve is added to compare the performance of the algorithm. In response to this comment, for the detailed revisions, please refer to Figures 9-20 in this revised draft for specific revisions.

Point 6: The section 3 “Fault Diagnosis model of TLSCA-SVM algorithm”, the core of the contribution, needs more explanations, interpretations and details: Why does the equation (10) take this form? How does the equation (13) improve the decision making?

Response 6: It has been modified. In response to this comment, for the detailed revisions, please refer to Equations 10 and 16 for specific revisions.

Reviewer 2 Report

Respected Editors and authors,

Greetings,

The manuscript is well structured and well argued. However, several rectifications and modifications are required to ensure that its quality stands up to this reputed journal.  

In general,

  • The authors have chosen the Sine-Cosine algorithm with support vector machine and transfer learning to realize a novel fault diagnosis classifier with higher accuracy and lower computational times which is the strong point of the manuscript.
  • The English language must be improved. There are several grammatical errors as one goes through the manuscript that requires rectification. Most of the sentences convey no proper meaning and could be off-putting to the readers and practitioners.
  • It is recommended that the authors incorporate the latest meta-heuristics and state-of-the-art advanced optimizers like the SHADE algorithms to demonstrate the superiority of the proposed method as comparison with the classical paradigms such as PSO, GA and SA may not provide the best competition.
  • SCA has also been accused of being a defective optimizer by ‘P. Niu, S. Niu, N. liu, and L. Chang, “The defect of the Grey Wolf optimization algorithm and its verification method,” Knowledge-Based Syst., vol. 171, pp. 37–43, 2019, doi: 10.1016/j.knosys.2019.01.018’. Refer to page 42 in the conclusion section. It is also required that the authors address these controversies over the choice of SCA besides there being several other such abstractly designed optimizers.

Section wise comments (wherever necessary) are as follows.

Section – 1

  • The first section introduces a basic outlook on various industrial fault diagnosis techniques techniques and a brief review of SCA is presented.
  • SCA from 2016 has a number of publications with several improved and modified versions. Additionally, a few review articles on the algorithm are also available. Hence, the review of SCA must summarize their views and provide a broader overview as to what makes SCA the optimal choice. Additionally, the controversy surrounding SCA of its derogatory performance with biased/shifted benchmarking functions must be given a valid explanation in this regard. Please refer and cite such sources wherever required.
  • Besides SCA and machine learning, a broader literature survey citing previous successful realizations of other meta-heuristics, hybridization and combinatorial variants for fault diagnosis is required.

Section – 2

  • Equations 1 and 2 are duplicates. Please rectify them.
  • The authors have stated that in Page 4, line 195 that “Through the above iteration, it can be guaranteed that the algorithm has the same probability in global search and local search, so that the algorithm itself can combine the benefits of both search modes itself to achieve better results”. However, since the search equations only incorporate the current member and the gbest, it can be argued that only local search is glorified over global search. The population diversity could be lowered drastically since the search bubble/or circle confines each particle to a fixed solution space and lowers its radius linearly over the iterations. It is essential that a critical analysis of the algorithm at hand is required to judge its merits and shortcomings.

Section – 4

  • In Line 312, provide necessary references to the literature from where the fault data was procured.
  • More details and explanation as to how the PCA techniques achieves the reduction in dimensionality can be provided.

Section – 5

  • In the Sub-Section 5.1.1, Page 14 the lines from 421 to 432 have been copied from sub-section 2.1, Page 4, lines 159 to 169. Please refrain from providing the same redundant data.
  • If the authors wish to provide evidence to the performance improvement of SCA for its exploratory-exploitative capabilities, they can utilize the latest benchmarking standards like CEC2020 benchmarking suite or CEC2019’s 100-digit competition as they help analyze the accuracy and robustness of meta-heuristics much more precisely.
  • The details of the algorithm-specific tuning parameters such as the inertial weight, social and cognitive components in PSO, mutation and crossover rates in GA etc. have not been provided. The optimization criteria, namely, the number of iterations, population size, number of function evaluations, computational complexities of the considered algorithms, parametric tuning settings for the algorithms have not been provided. A simple tabulation of these would suffice considering the fact that the reproduction of the results would be easier and a direct comparison is feasible by the other researchers in case if the same problem is taken up in the future.
  • In Tables 3 and 4, the statistical analysis of the results is missing. Authors must provide the details of the best, worst, average and the standard deviation of the accuracy percentages with respect to each algorithm.
  • The comparisons with more meta-heuristics (modern and state-of-the-art advanced) and a few variants of SCA from the recent literature can be provided to reinforce the superiority with the SCA-SVM classifier.

Section - 6

  • To keep things fair, a brief discussion of the demerits with the proposed technique should be provided.

Final Comments

The manuscript has the potential to be improved and requires major rectifications. With that being said, I wish the authors all the best on their endeavor to improve the quality of the manuscript.

Author Response

Response to Reviewer 2 Comments

We would like to thank to the reviewer for the time and effort dedicated on the review of our paper and the very helpful comments and suggestions. According to the comments of the Associate Editor, we have made corresponding revisions to improve the quality of the manuscript. Replies to each individual question are provided as following:

Point 1:SCA has also been accused of being a defective optimizer by ‘P. Niu, S. Niu, N. liu, and L. Chang, “The defect of the Grey Wolf optimization algorithm and its verification method,” Knowledge-Based Syst., vol. 171, pp. 37–43, 2019, doi: 10.1016/j.knosys.2019.01.018’. Refer to page 42 in the conclusion section. It is also required that the authors address these controversies over the choice of SCA besides there being several other such abstractly designed optimizers.

Response 1: “Firstly, the controversy about the SCA algorithm, where there may be a controversy about the local search of the optimization algorithm. Referring to the relevant literature, it is found that the local search of the optimization algorithm is controversial. It is not difficult to understand that the optimal solution can be guaranteed under the global search. However, in the local search, it is inevitable that the local optimal solution may appear in the local search, or the performance at a certain point is the best, and the farther away from the point, the worse the performance. In this paper, the parameters of SVM are optimized by using the same characteristics of local search and global search probability of SCA algorithm in optimizing parameters. The SCA-SVM classifier is obtained and a good classification effect is obtained. The classification algorithm of this paper is analyzed. This paper uses the SVM classifier as the main body for fault diagnosis. The SVM classifier itself has a good classification effect, and the difference between the important parameter’s penalty factor and kernel parameter will affect the classification effect of the SVM. The use of the SCA algorithm is to obtain appropriate parameters. The search method is randomly determined each time an optimal solution is found. That is to say, in the next search, both the local search and the global search are random, that is, the probability is the same. Each time the optimal solution is approximated, the approximation method is randomly determined. Such an optimization method can avoid local optimal solutions and shorten the optimization time. The classifier formed after finding suitable parameters has a good classification effect in fault classification, and the classification efficiency is improved.

Secondly, I have benefited a lot from the suggestions made by the reviewers. Referring to related articles, I found the advantages and disadvantages of different optimization algorithms. Some algorithms combine the properties of multiple algorithms for optimization. For example, the gray wolf algorithm is combined with the sine and cosine algorithm to optimize parameters. These questions are worthy of my continuous study in my future study.

Finally, because of the inspiration of the problem, the optimization parameters of SCA are considered. And a discussion worth mentioning has been added to the article, so that readers have a clearer understanding.”

In response to this comment, for the detailed revisions, please see the Conclusion (c) and Conclusion (d)on page 31 for specific revisions.

Point 2: The first section introduces a basic outlook on various industrial fault diagnosis techniques and a brief review of SCA is presented.

SCA from 2016 has a number of publications with several improved and modified versions. Additionally, a few review articles on the algorithm are also available. Hence, the review of SCA must summarize their views and provide a broader overview as to what makes SCA the optimal choice. Additionally, the controversy surrounding SCA of its derogatory performance with biased/shifted benchmarking functions must be given a valid explanation in this regard. Please refer and cite such sources wherever required.

Response 2: Thanks for this excellent suggestion. In the revised version, [20-21] are cited to raise a controversy about the optimization algorithm. And make a reasonable explanation in conjunction with this article. In response to this comment, for the detailed revisions, please see the fourth paragraph of section 1 on page 2 for details of the revision

Point 3: Equations 1 and 2 are duplicates. Please rectify them.

Response 3: It has been modified. In response to this comment, for the detailed revisions, please refer to Equations 1 and 2 for specific revisions.

Point 4: To keep things fair, a brief discussion of the demerits with the proposed technique should be provided.

Response 4: In response to this comment, for the detailed revisions, please see the fourth paragraph of section 1 on page 2 and Conclusion (c)on page 31 for specific revisions.

Point 5: In Line 312, provide necessary references to the literature from where the fault data was procured.

Response 5:  In the simulation experiment, considering the selectivity of the simulated circuit data, for more rigorous consideration, the faults of the Sallen-Key circuit and the CSTV circuit are set according to the literatures [38-42]. Please see the first paragraph of section 4 on page 9 for details in the revision.

Round 2

Reviewer 1 Report

The second version suits me. Thank you for the authors to answer appropriately most of my questions

Reviewer 2 Report

Authors have addressed all the comments satisfactorily. Now, I suggest that this paper is suitable for publication.